# A new characterization of the North Atlantic eddy-driven jet using 2-dimensional moment analysis

Jacob Perez[1], Amanda C. Maycock[2], Stephen D. Griffiths[3], Steven C. Hardiman[4], and Christine M. McKenna[2]

[1]Centre for Doctoral Training in Fluid Dynamics, University of Leeds, Leeds, UK
[2]School of Earth of Environment, University of Leeds, Leeds, UK
[3]School of Mathematics, University of Leeds, Leeds, UK
[4]Met Office Hadley Centre, Exeter, UK

**Correspondence:** Jacob Perez (scjp@leeds.ac.uk)

**Abstract.** We develop a novel technique for characterising the latitude ($\overline{\phi}$), tilt ($\alpha$) and intensity ($U_{\mathrm{mean}}$) of the North Atlantic eddy-driven jet using a feature identification method and two-dimensional moment analysis. Applying this technique to the ERA5 reanalysis, the distribution of the daily winter $\overline{\phi}$ is unimodal which is in contrast to the trimodal distribution of the daily Jet Latitude Index (JLI) described by Woollings et al. (2010). We show that our method exhibits a higher persistence than the JLI, casting doubt on the previous interpretations of the trimodal distribution as evidence for regime behaviour of the North Atlantic jet. It also explicitly and straightforwardly handles days where the jet is split. Although climatologically $\alpha$ is positive, indicating a tilt from south to east, around a fifth of winter days show negative $\alpha$. When plotted as a function of the North Atlantic Oscillation and East Atlantic pattern indices, there is a higher fraction of explained variance in the daily $\overline{\phi}$ within each quadrant of the phase space than is found for JLI, supporting the conclusion that $\overline{\phi}$ has smoother variations than JLI and has a closer relationship with these indices. Our method is simple, requiring only the daily 850 hPa zonal wind as input data, and diagnoses the jet in a more informative and robust way than other methods using low-level wind fields.

## 1 Introduction

Weather and climate variability in the Euro-Atlantic region is strongly mediated by changes in the position and intensity of the North Atlantic eddy-driven jet (EDJ). Characterising the behaviour of the EDJ across timescales has been the subject of many studies (e.g., Woollings et al., 2010, 2018; Parker et al., 2019; Simpson et al., 2019) and a variety of methods have been developed to diagnose EDJ features, ranging from very simple to more complex. The methods can be separated into two broad categories that represent their use of upper or lower tropospheric dynamical information.

Approaches based on upper tropospheric circulation generally use 3-dimensional (longitude, latitude, pressure) horizontal wind fields and identify local wind speed maxima using selected thresholds (e.g., Koch et al., 2006; Limbach et al., 2012; Manney et al., 2014). Spensberger et al. (2017) analyse quasi-horizontal wind shear on the dynamical tropopause to locate 'jet axes' and relate their characteristics to wave breaking and blocking events. A key issue for 'upper tropospheric methods' is their ability to distinguish the EDJ from the Subtropical Jet (STJ), which may require further criteria to distinguish jet features

(Spensberger et al., 2023). Upper tropospheric jet metrics are particularly useful for connecting the jet with synoptic systems (e.g., Spensberger et al., 2017) and elucidating the influence of diabatic processes on the jet (e.g., Auestad et al., 2024).

The second category of EDJ identification methods uses the lower tropospheric circulation. This is attractive because in principle the baroclinic structure of the STJ makes it more straightforward to identify the EDJ from low-level westerlies. However, a key issue for 'lower tropospheric' EDJ identification methods is the potential effect of surface topography on wind fields, which may create local wind features that are not eddy-driven (e.g., Greenland tip jets, see White et al. (2019)). Existing 'lower tropospheric' methods have typically applied zonal or sectoral averaging of low-level zonal winds to identify the latitude

of the maximum wind speed (e.g., Woollings et al., 2010; Ceppi et al., 2014). Barriopedro et al. (2022) added further constraints and statistical models to this framework to characterise the strength and tilt of the EDJ, in addition to its latitudinal position. Lower tropospheric metrics have been shown to have close links to large scale modes of variability such as the North Atlantic Oscillation (e.g., Barnes and Hartmann, 2010; Woollings et al., 2010) and the zonal momentum budget in the mid-latitudes (e.g., Simpson et al., 2014).

Another important consideration for both 'upper and lower tropospheric' EDJ identification methods is the choice of whether to time filter the input data. Some methods use instantaneous fields with the view of preserving a direct link to associated synoptic events (e.g., Manney et al., 2014; Spensberger et al., 2017), while other methods use low-pass filtered data to remove individual weather events and focus on slower variations in the background wind field (Woollings et al., 2010; Barriopedro et al., 2022).

Arguably, the most widely utilised method for EDJ identification is currently the Jet Latitude Index (JLI) of Woollings et al. (2010). The JLI is defined as the latitude of the maximum North Atlantic sector-averaged low-pass filtered lower tropospheric zonal wind. In boreal winter, the daily JLI exhibits a trimodal distribution in the North Atlantic basin (Woollings et al., 2010), which has been interpreted as the jet exhibiting preferred states or regime behaviour (e.g., Frame et al., 2011; Franzke et al., 2011; Novak et al., 2015). The trimodal behaviour of the JLI has been linked to many phenomena, including storm track

variability (Novak et al., 2015), phases of the North Atlantic Oscillation (NAO) (Woollings and Blackburn, 2012), North Atlantic weather regimes (Madonna et al., 2017), state-dependent weather predictability (Frame et al., 2011) and stratosphere-troposphere dynamical coupling (Maycock et al., 2020). It is also used as a diagnostic for model evaluation (Simpson et al., 2020).

Despite its wide application, the JLI offers a simplified view of the EDJ and questions have been raised about the physical

interpretation of the trimodal distribution. For example, White et al. (2019) suggested that the northern JLI peak is primarily a result of Greenland tip jets and is not a manifestation of the EDJ. Furthermore, it is unclear how the JLI performs for zonal wind profiles that are bimodal (e.g., Figure 12 of Woollings and Blackburn, 2012). A key feature of the North Atlantic EDJ is its distinct SW-NE tilt due to stationary wave forcing by orography and sea surface temperature patterns (Brayshaw et al., 2009). However, because of zonal averaging over the North Atlantic basin, the JLI does not explicitly account for jet tilt.

Some studies have proposed methods to calculate the tilt of the North Atlantic jet, such as Messori and Caballero (2015) who identify the latitude of the maximum wind speed at each meridian and calculate a Jet Angle Index (JAI) from the line of best fit through the maxima (for a similar approach using zonal wind see Barriopedro et al., 2022). However, the latitude of the wind

speed maximum can show large jumps between the meridians, particularly on days when the jet is split, and such spurious behaviour needs to be filtered out using arbitrary criteria (Messori and Caballero, 2015). Barriopedro et al. (2022) also extend the measures of the EDJ to better characterise its variability by introducing measures of longitudinal position and sharpness; however, the implementation of their measures relies on similar underpinning assumptions as Woollings et al. (2010) and Messori and Caballero (2015).

The goal of this study is to offer a more robust, yet relatively simple, approach for characterising the North Atlantic EDJ in the lower troposphere. To achieve this, we apply moment analysis to two-dimensional jet objects identified using a feature-based approach. Moment analysis has been applied to study other dynamical phenomena of geophysical fluids, including the Stratospheric Polar Vortex (e.g., Waugh and Randel, 1999; Matthewman et al., 2009) and the nature of coherent vortices in turbulent flows (Mak et al., 2017). This approach has the advantage that it does not rely on spatially-averaged input data and the diagnostics are defined by a collection of points over a specified region of interest rather than a single point. Although similar area-weighted definitions of EDJ position and strength have been used (e.g., Woollings and Blackburn, 2012; Ceppi et al., 2014), they have been based on one-dimensional wind profiles. Using two-dimensional moment analysis, we define the position, tilt, and the strength of the EDJ in a manner that is simpler and more intelligible than other proposed 'lower tropospheric' EDJ methods (e.g., Barriopedro et al., 2022). Another benefit of the method compared to the JLI is that it allows for days where there is no well-defined EDJ and can readily accommodate days where the jet is split. Further, issues with the identification of orographic features are mitigated by incorporating a minimum EDJ length, which is shown to reduce the occurrence of jets located at high northern latitudes. While the focus of this study is on the North Atlantic region in winter, the methods are general and could be applied to other regions or seasons.

The paper is structured as follows. Section 2 outlines our new methodology for characterising the North Atlantic EDJ. Section 3 compares the new diagnostics with the JLI and the JAI. Section 4 relates the new diagnostics to the two leading modes of North Atlantic atmospheric variability, i.e., the North Atlantic Oscillation (NAO) and the East Atlantic Pattern (EA). We finish in Section 5, with conclusions and recommendations.

## 2   Data and Methods

### 2.1   Data

We used daily average data for December-February from the ERA5 reanalysis dataset over 1979-2020 (Hersbach et al., 2020). All fields are bilinearly interpolated to a regular $2° \times 2°$ grid.

### 2.2   Identification of EDJ objects

EDJ objects are identified using 850 hPa zonal wind ($U_{850}$) in the same North Atlantic domain ($15°-75°$N, $0°-60°$W) used in the JLI calculation (Woollings et al., 2010). A single wind level is used as there were only minor differences in the results when vertically averaging between 725-950 hPa as is done in Woollings et al. (2010). Following Woollings et al. (2010), a

low-pass Lanczos filter with a 61-day window and a 10-day cut-off frequency is applied to remove short time-scale synoptic features (Duchon, 1979). We tested the sensitivity of the results to omitting the time-filtering of the wind inputs and find there are minor differences in the overall jet statistics presented in Section 3 (see Section A1 of the Appendix and Figure A1 for details). After applying the time filter and accounting for the complete winter seasons, this leaves a total of 3701 days in our analysis.

The identification of an Eddy-Driven Jet Object (EDJO) is outlined by the flowchart shown in Figure 1. Implementing the algorithm requires the choice of the three parameters: a wind speed threshold $U_{850}^*$, a minimum geodesic length threshold $L^*$, and a minimum zonal extent threshold $L_\lambda^*$. The steps of the algorithm are as follows:

1. **Locate seed points** - Seed points are identified as local maxima in the $U_{850}$ field, denoted $U_{\max}$. Note that multiple seed points may be identified in an image. If $U_{\max} < U_{850}^*$, then no EDJO is found for that day and the remainder of the steps are skipped.

2. **Flooding** - Starting from the seed point with the largest wind maximum, all neighbouring grid points where $U_{850} \geq U_{850}^*$ are recursively tagged, and a contour enveloping these points is defined as the EDJO (see contour in Figure 1). The neighbouring points include those that are above, below, left, right, and diagonally adjacent to the seed point.

3. **Length check** - A key feature of EDJs is that they are large-scale zonal jets. To remove small-scale local wind features, we apply two length checks to the EDJO identified in Step 2. First, we require the geodesic length of an EDJO, $L$, to satisfy $L \geq L^*$. To calculate $L$, a line is extrapolated through the centre of mass of the object (blue circle in Figure 2) and along the main axis (longer black line coming out of the centre of mass in Figure 2), using the distance between the two points that intersect the edge of the EDJO. The definitions of centre of mass and major axis are given in Section 2.3.

   Second, we require that the EDJO extends over a minimum longitudinal extent, so that the longitude range spanned by the EDJO must satisfy $L_\lambda \geq L_\lambda^*$. If either of these length checks is not met, then the EDJO is rejected.

4. **Remaining Maxima** - To avoid duplicating EDJOs (e.g., if more than 1 maximum lies within a single EDJO), the associated grid points from the previous EDJO are removed from the $U_{850}$ field. If there are any other remaining seed points, then the algorithm returns to step 2 and repeats. This is an advantage over previous methods as it enables the characteristics of split jets to be retained. Once there are no remaining $U_{850}$ maxima, the algorithm moves to the next time step.

## 2.3 Moment Diagnostics

Moments are common in statistics to define properties of a distribution, such as the mean or variance. The definition used here is

$$M_{pq} = \int \int_\Omega \lambda^p \phi^q U_{850}(\lambda, \phi) \ r^2 \sin\phi \, d\lambda d\phi, \tag{1}$$

where $\Omega$ is the EDJO, $\lambda$ and $\phi$ are the longitude and latitude, respectively, and $r^2 \sin \phi d\lambda d\phi$ is the area element on a sphere where $r$ is the radius of the Earth. This formulation is similar to that applied to the potential vorticity distribution for studying the stratospheric polar vortex (e.g., Waugh, 1997). A key difference is that we have chosen to include the strength of the zonal wind as a weighting in the calculation. The inclusion of the weighting factor $U_{850}$ in equation (1) means that our moment calculations (of position and tilt) will reflect regions of stronger zonal wind within the EDJO, which is important in the context of surface impacts. The lack of weighting results in purely geometrical moments, as used in some previous studies (e.g., Waugh, 1997; Waugh and Randel, 1999).

The weighting factor also allows us to calculate quantities that are analogues of those used to quantify planar objects in mechanics (mass, centre of mass, major and minor axes), where the weighting factor is simply the surface density (i.e., mass per unit area). Thus, we define the 'mass' of an EDJO to be $U_{\mathrm{mass}} = M_{00}$, with units of $\mathrm{m}^3\mathrm{s}^{-1}$. The average jet strength, $U_{\mathrm{mean}}$, with units of $\mathrm{m}\,\mathrm{s}^{-1}$, is then

$$U_{\mathrm{mean}} = \frac{U_{\mathrm{mass}}}{\int \int_\Omega r^2 \sin \phi d\lambda d\phi}, \tag{2}$$

which is analogous to the average surface density in planar mechanics. Most of the results in this paper represent the largest mass EDJO on a given day, but complete statistics of all EDJOs are provided in the supporting information. The jet position can be described by the analogue of the centre of mass, which arises as a longitude $\overline{\lambda}$ and latitude $\overline{\phi}$ defined by

$$\overline{\lambda} = \frac{M_{10}}{M_{00}}, \quad \overline{\phi} = \frac{M_{01}}{M_{00}}. \tag{3}$$

The jet orientation is described by the major axis, which requires the analysis of the analogue of the inertia matrix I, here defined by

$$\mathsf{I} = \begin{pmatrix} \tilde{M}_{02} & -\tilde{M}_{11} \\ -\tilde{M}_{11} & \tilde{M}_{20} \end{pmatrix}, \quad \text{where} \quad \tilde{M}_{pq} = \int \int_\Omega (\lambda - \overline{\lambda})^p (\phi - \overline{\phi})^q U_{850}(\lambda, \phi) \, r^2 \sin \phi d\lambda d\phi. \tag{4}$$

The main axis of the EDJO (the longer black line coming from the blue dot in Figures 2a and c) is the direction of the eigenvector associated with the lower eigenvalue of I. We define the jet tilt, $\alpha$, as the angle between the major axis and the latitude line $\phi = \overline{\phi}$, with positive values indicating a SW-NE tilt and vice versa. For EDJOs with $\tilde{M}_{20} > \tilde{M}_{02}$, that is, those elongated longitudinally rather than latitudinally, as should be confirmed by the length checks in Step 3 of Section 2.2, there is a simple expression for $\alpha$:

$$\alpha = \frac{1}{2} \arctan \left( \frac{2\tilde{M}_{11}}{\tilde{M}_{20} - \tilde{M}_{02}} \right), \tag{5}$$

as also used in Matthewman et al. (2009).

## 2.4 Choice of $U^*_{850}$, $L^*$ and $L^*_\lambda$

To focus on eddy-driven westerly jets, we set $U^*_{850} = 8\,\mathrm{ms}^{-1}$ for this study. This value has been used in other studies of the winter North Atlantic EDJ (e.g., Woollings et al., 2010). We find that the overall statistics of $\overline{\phi}$ and $\alpha$ are not very sensitive to

the choice of $U_{850}^*$ for values between 6 and $11\,\mathrm{ms}^{-1}$; however, the median value of $U_{\mathrm{mean}}$ scales with $U_{850}^*$ as expected (see Appendix A2 and Figure A2 for details). We note that to identify jets in other regions and in other seasons, a different value of $U_{850}^*$ may be more suitable.

The inclusion of minimum thresholds $L^*$ and $L_\lambda^*$ is to remove small-scale westerly features that are not related to the EDJ. The precise value of $L^*$ is somewhat arbitrary, but after testing, we set $L^* = 1661\,\mathrm{km}$, which is the geodesic length of a purely zonal jet at the most northerly latitude in our domain (75°N). Note that this is larger than the Rossby radius of deformation in the mid-latitudes of around $1000\,\mathrm{km}$, so it is sufficient to remove most synoptic-scale wind features even in unfiltered data (e.g., jet streaks within extratropical cyclones).

Although $L^*$ is sufficient to remove many small-scale objects, we found some low-latitude features with a sufficient meridional extent to satisfy $L > L^*$, but which do not resemble an EDJ. We therefore introduce $L_\lambda^* = 20°$ to ensure that the EDJOs have a considerable zonal extent as expected for an EDJ. The effect of including $L^*$ and $L_\lambda^*$ thresholds in the algorithm is shown in Section A3 of the Appendix. The main effect of $L^*$ and $L_\lambda^*$ is to remove small objects with $\overline{\phi} > 65°$N (see Appendix A3 and Figure A3). These objects coincide with the northerly JLI mode, indicating that including length constraints in the algorithm removes a substantial fraction of the very high northerly latitude wind features likely associated with orographic effects (White et al., 2019).

## 2.5   JLI and JAI methodologies

The JLI is calculated using the method described by Woollings et al. (2010), but applied to the $850\,\mathrm{hPa}$ zonal wind field for consistency. Briefly, it is the latitude of the maximum of the north Atlantic sector averaged $U_{850}$, where the maximum is found from a third-order polynomial to allow maxima to be found between the grid points. The data are time-filtered using the same Lanczos filter as above. The jet speed ($\mathrm{JLI_{vel}}$) is the zonal wind in the JLI.

The JAI is calculated as in Messori and Caballero (2015). It uses the daily filtered wind speed at $850\,\mathrm{hPa}$ ($\mathrm{WS_{850}} = \sqrt{U_{850}^2 + V_{850}^2}$) and locates the maximum at each meridian across the basin. A meridian is ignored when there is a secondary maximum within $4\,\mathrm{m\,s}^{-1}$ of the largest maximum and further than 5° in latitude away. This screening is applied to ignore any split jets in the fitting. Linear regression is applied to the maxima across the basin to give a line of best fit. With this the JAI is an angle between -180° and 180° that can be calculated from one of the (overlapping) definitions

$$
\mathrm{JAI} = \mathrm{atan2}(\Phi_1 - \Phi_0, \Lambda_1 - \Lambda_0) = \begin{cases} \arctan(\frac{\Phi_1 - \Phi_0}{\Lambda_1 - \Lambda_0}) & \text{if } \Lambda_1 - \Lambda_0 > 0, \\ 90° - \arctan(\frac{\Lambda_1 - \Lambda_0}{\Phi_1 - \Phi_0}) & \text{if } \Phi_1 - \Phi_0 > 0, \\ -90° - \arctan(\frac{\Lambda_1 - \Lambda_0}{\Phi_1 - \Phi_0}) & \text{if } \Phi_1 - \Phi_0 < 0, \\ \arctan(\frac{\Phi_1 - \Phi_0}{\Lambda_1 - \Lambda_0}) \pm 180° & \text{if } \Lambda_1 - \Lambda_0 < 0, \\ \text{undefined} & \text{if } \Lambda_1 - \Lambda_0 = 0 \text{ and } \Phi_1 - \Phi_0 = 0, \end{cases} \tag{6}
$$

where $(\Phi_0, \Lambda_0)$ and $(\Phi_1, \Lambda_1)$ are the start and end points of the best fit line. Note that this method does not include information on wind direction, only the wind speed. The JLI and JAI methodologies only assign a single jet latitude and tilt for each day,

respectively. This contrasts to our two-dimensional moment diagnostics, which assign values of $(\overline{\lambda}, \overline{\phi})$ and $\alpha$ for each EDJO identified on a given day.

## 2.6 North Atlantic modes of atmospheric variability

Deseasonalised and standardised daily mean sea level pressure (MSLP) fields are used to calculate the North Atlantic Oscillation (NAO) and the East Atlantic pattern (EA) indices. Following Baker et al. (2018) and McKenna and Maycock (2021), the NAO index is defined as the pressure difference between the grid boxes closest to Gibraltar $(36°\text{N}, 5.3°\text{W})$ and Iceland $(65°\text{N}, 22.8°\text{W})$. The East Atlantic index is defined as the MSLP anomaly in the gridbox closest to $52°\text{N}, 27.5°\text{W}$ (Moore et al., 2011).

## 3 Results

### 3.1 Case study comparisons with JLI and JAI

Figure 2 illustrates two example days (rows) where the moment analysis, JLI and JAI methods are applied. The left column shows $U_{850}$ from which the moment analysis and JLI are calculated, and the right column shows $\text{WS}_{850}$ from which JAI is calculated. On the first example day (Figure 2a and b), there is close agreement between $\overline{\phi}$ and the JLI (Figure 2a). However, there is strong disagreement between $\alpha$ and JAI, which show opposite signs (Figure 2b). This is because the JAI calculation does not account for the direction of the wind and erroneously connects points with intense zonal westerly and east wind (Figure 2 a), leading to a positive JAI that does not characterize the tilt of the westerly jet on this day. Furthermore, the meridians are only sparsely sampled because of the JAI criterion to disregard split jets. Therefore, the JAI calculation can result in inaccurate assessments of tilt for several reasons.

In the second example (Figures 2c and d), the JAI and $\alpha$ are in close agreement showing a highly tilted jet structure (Figure 2d). However, there is a marked difference between the JLI and $\overline{\phi}$ on this day (Figure 2c). This discrepancy is attributable to the specific EDJ structure comprising a highly tilted jet with an intensified equatorward flank. As a result of the stronger equatorward westerlies, the JLI shows a southerly position. In contrast, $\overline{\phi}$ exhibits a more central latitude that better encapsulates the overall structure of the jet on this day. These two examples illustrate that JLI and JAI can provide misleading results and that moment-based analysis has the potential to overcome some of their respective limitations.

We now consider the temporal variability of the different diagnostics of EDJ. During a single winter, the JLI can display large changes in a short period (Woollings et al., 2010; Madonna et al., 2017). These changes have been interpreted as regime changes, with sudden transitions between "preferred" jet latitudes (Hannachi et al., 2011; Franzke et al., 2011; Novak et al., 2015). Figure 3 shows the evolution of the JLI and $\overline{\phi}$ for the winter of 2016/17. During some periods, JLI and $\overline{\phi}$ show similar jet positions. However, at other times, the JLI displays large rapid changes that are not consistently mirrored in $\overline{\phi}$. Figure 3 also shows that at certain times the moment-based method identifies a split jet (pink markers) that implicitly cannot be captured by the JLI algorithm. In general, $\overline{\phi}$ exhibits a more smoothly varying temporal behaviour during the season.

To examine what is happening during some of the periods in 2016/17 where the two measures of jet latitude show different behaviours, we select three snapshots from the winter (black arrows in Figure 3) and plot the $U_{850}$ maps for these consecutive days (Figure 4). The 10th and 11th December 2016 coincide with an 8° decrease in JLI but little change in $\overline{\phi}$. Figures 4a and 4b reveal a broad EDJ in this period. When the EDJ is broad, the JLI is very sensitive to relatively small fluctuations in the zonal wind field within the EDJO. However, $\overline{\phi}$ is less sensitive to local fluctuations and varies relatively little between these days, which better captures the fact that the overall EDJO is largely unchanged.

In early January 2017, an atmospheric blocking event in the North Atlantic resulted in a split EDJ. During this period, the JLI shows a northerly jet, but this does not well characterise the overall circulation pattern (Figures 4c and 4d). In contrast, the moment-based method locates two separate EDJOs which better reflect the split jet structure.

Between January 22 and 23 2017, the JLI shows a large north-to-south transition of around 25° in a single day, while $\overline{\phi}$ shows almost no change in latitude. Figures 4e and 4f show that the jet is highly tilted during this period. Initially, the JLI locates the maximum wind at a northern latitude near the tip of Greenland. Subsequently, a very modest strengthening in the westerlies over West Africa causes a large equatorward shift in the JLI the following day. Remarkably, the large-scale $U_{850}$ field remains relatively unchanged during this 'transition', and this is reflected not only in $\overline{\phi}$, but also in the value of $\alpha$ in its respective time series (not shown).

These three examples illustrate the limitations of JLI as a measure of the jet. We conclude that JLI may provide misleading results in the following cases: 1) a broad jet, 2) a highly tilted jet, 3) a split jet, and 4) when there is no well-defined EDJ. In all these cases, our moment-based method offers a more detailed and informative picture of the overall jet structure. To further illustrate this, Figure 5 shows a time series of the frequency of large changes in jet latitude based on JLI and $\overline{\phi}$. A large change is defined as a change in latitude $\geq 10°$ between consecutive days. For $\overline{\phi}$, this is calculated only on the days when a single EDJO is defined, to reduce the potential switching between the largest mass object on consecutive days with two EDJOs. As shown above, days with a single EDJO are less sensitive to broad or tilted jet cases. The mean winter frequency of large shifts in the JLI is 7 times greater than for $\overline{\phi}$. This indicates that the characteristic of the JLI that exhibits a sudden change, which was highlighted in the example winter in Figure 4, is generally representative of other years.

## 3.2 Winter statistics

The distributions of $\overline{\phi}$ and JLI on all days are shown in Figures 6a and b. Equivalent distributions for all EDJOs are shown in Supplementary Information Figure S1. The two measures show comparable value ranges but different structures. Note that $\overline{\phi}$ has a (roughly Gaussian) unimodal distribution, with mean $\mu(\overline{\phi}) = 45.7°$. There is a slight negative skew of -0.07±0.09 (95% confidence interval), which is particularly evident on the equatorward flank near 37°. As noted in many previous studies, JLI shows a trimodal distribution with maxima near 37°N, 45°N and 57°N. The distribution of differences between $\overline{\phi}$ and JLI binned by the value of the JLI (Figure 6c) reveals that when the JLI is in the southern mode (JLI $< 40°$; 12.1% of days), $\overline{\phi}$ tends to take more poleward values than the JLI with a median difference of 0.74° (and standard deviation $\sigma$=3.7°). When the JLI is located in the northern mode (JLI $> 52°$; 37.4% of days), $\overline{\phi}$ tends to be more equatorward than the JLI with a median value of 5.2° ($\sigma$=7.23°). When the JLI is in the central mode ($40° \leq$ JLI $\leq 52°$; 48.5% days), the two methods show the highest

agreement with a median difference of -1.0° and a standard deviation of 2.28°. Therefore, fundamental differences in the shape of the distributions arise predominantly from days when the JLI is in the northern or southern modes, which arises due to the area-weighted definition of the position. The distribution of $\overline{\phi}$ including all EDJOs, and not just the largest mass one, is very similar and also does not show multi-modal structure (Supplementary Figure S1a).

The distributions of $\alpha$ and JAI in Figures 6d and 6e show a mean positive tilt for both measures ($\mu(\alpha)$=7.9°, $\mu$(JAI) = 11.7°), consistent with the winter climatology of $U_{850}$. The JAI exhibits higher variability than $\alpha$ with $\sigma = 15.7°$, while $\alpha$ has a value of 11.7°. The median $\alpha$-JAI difference is -3.6° when JAI$\geq$0 and 10.9° when JAI<0, indicating that JAI tends to produce larger magnitudes of tilt than $\alpha$ (Figure 6f). The differences in $\alpha$ -JAI also show a large spread, with $\sigma$=13.4° for JAI$\geq$0 and $\sigma$=13.7° for JAI<0, indicating many outlier days where the two tilt measures are substantially different. Some of these days

are associated with opposite signs of $\alpha$ and JAI due to the fact that JAI uses the wind speed without accounting for the wind direction.

     It was previously shown in Section 3.1 that strongly tilted jets can cause disparities between $\overline{\phi}$ and JLI. To further investigate this, Figure 7 shows composites of $U_{850}$ binned by $\alpha$ where $\alpha \geq 0$, with the values of $\overline{\phi}$ and JLI overplotted as calculated from the (single) composite field of $U_{850}$. As $\alpha$ increases, the difference between the JLI and $\overline{\phi}$ also increases, with the JLI

identifying progressively more northerly values than $\overline{\phi}$ on average. The composite $U_{850}$ field for $\alpha \geq 30°$ (Figure 7c) shows two wind maxima near 30°N, 60°W and 62°N, 10°W, which suggests an influence from split jet days when $\alpha$ is strongly positive and also potentially Greenland tip jet days (White et al., 2019). This shows that the substantial differences between $\overline{\phi}$ and JLI when JLI$\geq$52°N (Figure 6c) are linked to highly tilted jets, similar to the case shown in Figure 4c. This is further evidenced by the distribution of daily $\overline{\phi}$ − JLI differences with points coloured by $\alpha$ and $U_{\text{mean}}$ (Supplementary Figure S2). On

average, the differences between $\overline{\phi}$ and JLI are greatest when the jet is weaker and more tilted.

### 3.3   Temporal relationships

     The winter case study shown in Figure 3 showed weaker time variability in $\overline{\phi}$ than JLI. We now examine the overall persistence of the jet diagnostics. The autocorrelation function (ACF) for each of the JLI and $\overline{\phi}$ is shown in Figure 8a with shading showing two standard errors around the mean. There is a systematically higher ACF of $\overline{\phi}$ for lags of up to 10 days, which is

265 consistent with the emerging picture of a more noisy behaviour in JLI which would tend to reduce the ACF. The ACFs for $\alpha$ and JAI are similar. Interestingly, JLI$_{\text{vel}}$ has a higher ACF than $U_{\text{mean}}$ and $U_{\text{max}}$, which is the closest measure to JLI$_{\text{vel}}$. The lower ACF in $U_{\text{max}}$ may be caused by the search over the two-dimensional field, which may inherently cause some noise. The lower ACF in $U_{\text{mean}}$ is surprising as it is a mean over the EDJO, whereas JLI$_{\text{vel}}$ is the maximum at a single point. Sectoral averaging may smooth out some of the noise in speed, resulting in a higher ACF in JLI$_{\text{vel}}$. The highest ACF for the strength

comes from $U_{\text{mass}}$, which is interesting, since $U_{\text{mass}}$ and $U_{\text{mean}}$ are related simply by the area of the EDJO (see Equation 2).

## 3.4 Relationship between EDJO metrics

The analysis of all selected years finds 1.7% days without an EDJO, 93.5% days with one EDJO, and 4.8% days with two EDJO. Unless otherwise stated, the results shown here are characteristics based on the largest mass object on any given day. The locations of the centres of mass of the EDJOs are shown in Figure 9. The spatial distribution of all centres of mass (Figure 9a) shows a trimodal structure, with a high concentration of points in the centre of the basin and two smaller density maxima located in the northeast and southwest quadrants of the basin. These outlier regions coincide with the majority of the locations of two EDJO days (blue points), indicating sites where split jets occur. The distribution of the centres of mass for the largest EDJO mass on each day (Figure 9b) is largely similar to Figure 9a, except that the north-east secondary maximum decreases, indicating that these EDJOs are generally lower in mass than the south-west located EDJOs on split jet days. This is consistent with the smaller area of the high-latitude EDJOs.

To better understand the differences between the JLI and the distribution of the centre of mass, Figure 10 shows the locations of the EDJO centres of mass for days corresponding to the three JLI peaks. This shows that the centres of mass generally lie within the ranges of JLI for the southern and central modes (Figures 10a and b). However, for the northern JLI mode (Figure 10c) this pattern breaks down, and there is a much larger spread of centres of mass with many points lying outside the JLI range. The northerly JLI mode also coincides with the highest percentage of days with two EDJOs (2.8% days), followed by the southern JLI mode (1.4% days) and the central JLI mode (0.6% days). The northern JLI mode also contains, on average, EDJOs with the highest average $\alpha$ with $9.9°$; the central and southern modes display more zonal jets with an average $\alpha$ of $0.7°$ and $1.4°$, respectively.

We now investigate the cross-relationships between some of the jet parameters. Figure 11 displays scatterplots that illustrate the relationships between $\overline{\phi}$, $\alpha$, and $U_{\mathrm{mean}}$. There is little to no correlation between $\overline{\phi}$ and both $\alpha$ and $U_{\mathrm{mean}}$, with a Pearson correlation coefficient of 0.1 in each case (Figures 11a and 11b). However, in Figure 11c, a noticeable cone-shaped relationship is observed between $U_{\mathrm{mean}}$ and $\alpha$, indicating that a stronger North Atlantic EDJ tends to be more zonal. Referring back to Figure 11b, the strongest zonal jets are located close to the mean of $\overline{\phi}$. As the tilt increases while moving poleward, the strength of the jet tends to decrease. This behaviour is consistent with previous studies that have investigated the relationship between jet latitude and strength (Woollings et al., 2018).

## 4 Relationship to large-scale modes of variability

The North Atlantic Oscillation (NAO) and the East Atlantic pattern (EA) are the two leading modes of atmospheric variability in the North Atlantic sector. Although these modes do not explain the full spectrum of EDJ variability, they are associated with substantial variations in jet latitude and strength (Woollings et al., 2010).

Figure 12 shows scatterplots of daily NAO and EA indices colored by $\overline{\phi}$ JLI (Figures 12a and b), $\alpha$ and JAI (Figures 12c and d) $U_{\mathrm{mean}}$, $\mathrm{JLI}_{\mathrm{vel}}$ (Figures 12e and f) and $U_{\mathrm{mass}}$ (Figure 12g). The mean $\mu$, standard deviation $\sigma$ and variance explained $(R^2)$ for each quadrant are given in brackets. Figure 12b reproduces the finding of Woollings et al. (2010) that jet latitude increases moving clockwise around the phase space, with a discontinuity between the highest and lowest JLI values occurring within the

NAO-/EA- quadrant. The discontinuity in the NAO-/EA- quadrant is associated with weak European/Scandinavian blocking and jets, where the jet diverts to the north or south around the block (see Figure 4a in Madonna et al., 2017). The equivalent plot for $\overline{\phi}$ shows similar behaviour, but the variance within each quadrant is smaller, resulting in a more smoothly varying distribution. The total $R^2$ for the daily $\overline{\phi}$ variance explained by the NAO and EA indices is 45%, which can be compared with 40% for JLI. The $R^2$ for $\overline{\phi}$ within each of the NAO/EA quadrants is systematically higher than for JLI (Figures 12a and 12b), with the largest difference between $\overline{\phi}$ and JLI being 12% for the NAO+/EA+ quadrant.

The higher variance of JAI compared to $\alpha$ is evident in Figures 12e and 12e. The broad pattern for both measures is positive or weakly negative angles for NAO+. For NAO- there are some differences, as $\alpha$ shows stronger negative or weakly positive angles, while JAI has large positive angles for NAO-. This may be related to JAI performance during blocked days. However, the variance in jet angle explained by the NAO and EA indices in each quadrant is considerably smaller than for jet latitude and strength, indicating that there is not a strong relationship between jet tilt and these modes of variability.

The patterns of $U_{\mathrm{mean}}$ and $\mathrm{JLI}_{\mathrm{vel}}$ in the NAO/EA phase space are very similar, although the range of values is smaller for $U_{\mathrm{mean}}$. The strongest jets span the outer envelope of the distribution across the largest EA+ and NAO+ points. Conversely, the weakest jets are confined to NAO-/EA- which often coincide with blocking. The JAI and $\alpha$ values show a more isotropic distribution within the phase space, but there is substantially higher variability in JAI than in $\alpha$, as shown in Figure 6. For $\alpha$, generally the stronger NAO- states coincide with more negatively tilted EDJs, while NAO+ is associated with positive tilt.

Unlike what was seen for jet latitude, there is no systematic difference in the total $R^2$ for jet strength, with $U_{\mathrm{mean}}$ showing slightly higher $R^2$ for EA+ and vice versa for $\mathrm{JLI}_{\mathrm{vel}}$ in the EA- quadrants. The last measure of jet strength is $U_{\mathrm{mass}}$, which incorporates the EDJO area (Figure 12g). $U_{\mathrm{mass}}$ has a slightly different pattern in the NAO/EA phase space compared to $U_{\mathrm{mean}}$. On average, the lowest $U_{\mathrm{mass}}$ values occur in the NAO-/EA- quadrant and both NAO-/EA+ and NAO+/EA+ coincide with higher $U_{\mathrm{mass}}$. The correlation coefficient between $U_{\mathrm{mass}}$ and the EA index is 0.65, which can be compared to a correlation of 0.21 with the NAO index. In addition to this, $U_{\mathrm{mass}}$ shows higher $R^2$ in all quadrants than $U_{\mathrm{mean}}$ and $\mathrm{JLI}_{\mathrm{vel}}$. Hence, $U_{\mathrm{mass}}$ offers an alternative measure of EDJ strength that is closely related to the pulsing variability associated with EA.

## 4.1 Two Jet Object Days

Days when two EDJOs are identified correspond to a split EDJ. Although the climatological occurrence of two EDJO days is 4.9%, we find that up to 15% days can show a split jet within a single winter. The composite $U_{850}$ for two EDJO days is shown in Figure 13a. This shows regions of strong westerlies centred over the east coast of the USA and near Iceland, with a region of weak easterlies near the Bay of Biscay and to the west of Portugal. This pattern resembles the circulation during Atlantic blocking. Consequently, there is also a link between the occurrence of two EDJO days and the NAO/EA with most days coinciding with NAO-/EA- conditions (Figures 13b and 13c). In general on two EDJO days, the largest $U_{\mathrm{mass}}$ EDJO is located at lower latitudes south of 40°N (Figure 13b), while the lower $U_{\mathrm{mass}}$ EDJO is typically located at higher latitudes north of 50°N (Figure 13c); this is consistent with the stronger $U_{850}$ anomalies at lower latitudes in Figure 13a and the tendency towards smaller area EDJOs with increasing latitude.

## 5  Discussion and Conclusions

The Jet Latitude Index (JLI) of Woollings et al. (2010) is commonly used to characterise the location of the North Atlantic Eddy-Driven Jet (EDJ) in the lower troposphere (e.g., Woollings et al., 2018; Simpson et al., 2020; Maycock et al., 2020). The daily winter JLI values in the North Atlantic show a trimodal distribution, which has become a benchmark for theoretical and numerical models alike. A separate method, the Jet Angle Index (JAI) of Messori and Caballero (2015), has been proposed to diagnose jet tilt based on the lower tropospheric wind speed. Both of these methods have limitations, in part due to the use

of zonal averaging for the JLI and the use of wind speed, but not direction, for the JAI. To address these limitations, we have developed an intuitive and simple method that extracts Eddy-Driven Jet Objects (EDJOs) in the lower troposphere from a $U_{850}$ field. By calculating the two-dimensional moments of the EDJO, we define the position of the EDJ using an analogue of the centre of mass ($\overline{\lambda},\overline{\phi}$), its strength ($U_{mean}$) and tilt ($\alpha$). The same approach can also be used to extract information on jet width. Barriopedro et al. (2022) recently proposed a new multi-parametric method building on the JLI to characterise the structure of

the North Atlantic jet in more detail. However, most of the measures they consider can be derived from the moment analysis applied here and their method is considerably more complicated to implement. The method presented here is simple to apply to most observation and model datasets, requiring only daily average 850 hPa zonal wind fields, and for a given domain has only three tuneable parameters.

Our study has several key results:

1. Our jet identification method more robustly characterises the jet structure on days where the jet is broad, highly tilted, split or not well defined, as compared to the JLI.

2. The time variability of $\overline{\phi}$ shows fewer large amplitude 'jumps' between consecutive days as compared to JLI. Examination of cases suggests that these jumps in JLI can be spurious resulting from selecting one of several competing maximums and do not reflect meaningful changes in the jet structure. The autocorrelation function of $\overline{\phi}$ shows greater

persistence than JLI between days 2-9.

3. The statistics of $\overline{\phi}$ over all winter days do not show a trimodal distribution as seen for the JLI. The distribution has a mean of 45.7° and a skewness of -0.07. The daily differences between $\overline{\phi}$ and the JLI tend to be greater for larger jets and highly tilted jets. When compositing low-level zonal wind for the same range of values, the two measures pick out similar patterns of large-scale circulation.

4. There is a smoother variation of $\overline{\phi}$ and $U_{mean}$ in the NAO/EA phase space. Both $\overline{\phi}$ and $U_{mean}$ also have a higher variance explained by the NAO and EA indices than the JLI and JLI$_{vel}$.

5. The distribution of jet tilt $\alpha$ is more Gaussian with a mean of 7.9°. Around 20% of the days show a negative tilt.

The results presented here suggest that previous studies that analyse the time variability of the JLI and its connection to dynamical processes should be revisited (e.g., Novak et al., 2015; Franzke et al., 2011). Furthermore, past work to connect the

trimodal structure of the JLI to weather types or weather regimes should be revisited in light of the unimodal structure of $\overline{\phi}$

(Madonna et al., 2017). Future work will examine the impact of teleconnections on the North Atlantic jet, such as from El Niño Southern Oscillation (Hardiman et al., 2019) and the Indian Ocean Dipole (Hardiman et al., 2020), as well as the jet response to external forcings. This study has focused on the North Atlantic basin in winter, but the method could be equally applied to other regions and seasons.

It is important to note that our study only addresses 'lower tropospheric' EDJ identification methods. One limitation affecting lower tropospheric jet metrics is that different approaches are used for interpolating below the surface over high topography, e.g. Greenland, which can affect the input wind fields. Therefore, care is required to check the influence of, e.g. missing data over Greenland for lower tropospheric metrics. Another class of jet identification methods focus on the upper troposphere and have revealed important dynamical features related to jet behaviour (e.g., Limbach et al., 2012; Manney et al., 2014; Spensberger

et al., 2017). In a future study, it would be interesting to investigate the relationship between our lower tropospheric diagnostics and existing upper tropospheric methods. We encourage the community to adopt this as a new approach to characterising the lower tropospheric North Atlantic EDJ.

*Code availability.* The EDJO identification algorithm can be found at https://github.com/scjpleeds/EDJO-identification. Data used to produce the for analysis and to produce the figures in this work is available on Zenodo at 10.5281/zenodo.10053895. ERA5 data was downloaded

and processed from the C3S archive.

*Data availability.* The ERA5 dataset was downloaded from the C3S Copernicus Climate Data Store.

## Appendix A: Robustness of EDJO identification

### A1    The effect of time-filtering

Here we show the dependence of the main results of the study on the use of the low-pass Lanczos filter on the input data.

Figure A1 shows comparisons of $\overline{\phi}$, $\alpha$ and the number of EDJOs per day for filtered and unfiltered $U_{850}^*$. There are very modest differences in $\overline{\phi}$ and $\alpha$ (Figures A1a and b), with each distribution having a very similar mean ($\mu$) and a slightly larger standard deviation ($\sigma$) for unfiltered data. The number of EDJOs per day (Figure A1 c) shows a nearly identical number of zero EDJO days in both cases. There is a slightly higher frequency of two EDJO days in the unfiltered data, which can be attributed to the stronger winds, as this would allow for longer-lasting EDJOs. Overall, we conclude that time filtering has a negligible effect on

the detection of EDJOs and their resulting statistics. We note this result is likely to depend on the inclusion of explicit spatial filtering through $L^*$ and $L_\lambda^*$, which remove small EDJOs that might be more frequently detected in unfiltered data.

## A2  Sensitivity to choice of $U^*_{850}$

The results in the main text use $U^*_{850} = 8\,\mathrm{ms}^{-1}$ . Here we compare the results using different values of $U^*_{850}$. Figure A2 shows the distributions of $\overline{\phi}$, $\alpha$ and the number of EDJOs per day for $U^*_{850}$ varying between 6 and 11 $\mathrm{ms}^{-1}$. Both $\overline{\phi}$ and $\alpha$ are largely insensitive to the choice of $U^*_{850}$ (Figures A2a and b). There are changes in the median value of $U_{\mathrm{mean}}$ (not shown), but this would be expected since $U^*_{850}$ sets the minimum $U_{\mathrm{mean}}$. Finally, the distribution of the number of EDJOs per day shows little differences for $U^*_{850}$ between 6-9 $\mathrm{ms}^{-1}$. Higher values of $U^*_{850}$ lead to a decrease in days with one EDJO and an increase in days with zero, potentially because they no longer meet the length criteria. From these results, we conclude that the EDJO algorithm is largely insensitive to sensible variations in $U^*_{850}$.

## A3  Sensitivity to inclusion of $L^*$ and $L^*_\lambda$

The effect of including the $L^*$ and $L^*_\lambda$ criteria in the algorithm is shown in Figure A3, where the distributions of $\overline{\phi}$, EDJO area and number of EDJOs per day are shown. Including only the $L^*_\lambda$ check results in a higher $\overline{\phi}$ density at northern latitudes, which are smaller EDJOs in area because of Earth's curvature. Including only the $L^*$ check removes the EDJOs at high north latitudes but retains a higher density of small EDJOs that occur on the south flank of the $\overline{\phi}$ distribution. Each of the length checks on their own results in an increase in the frequency of days with two EDJOs, which are typically smaller than on days with a single EDJO. Not until both $L^*$ and $L^*_\lambda$ are used together do we see a decrease in the occurrence of the smallest EDJOs.

*Author contributions.* JP, ACM, SDG, CMM and SCH devised the methodology. JP implemented the methodology, performed all analysis, produced the figures and led the writing of the paper. ACM, SDG, CMM and SCH assisted with writing the paper.

*Competing interests.* There are no competing interests.

*Acknowledgements.* JP was supported by the EPSRC Fluids Centre for Doctoral Training at the University of Leeds. SCH was supported by the Met Office Hadley Centre Climate programme funded by BEIS and Defra. ACM and CMM acknowledge support from the EU H2020 CONSTRAIN project (Grant Agreement No. 820829). ACM was supported by the Leverhulme Trust. We also thank Tim Woollings and one anonymous reviewer for their constructive feedback on the paper.

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

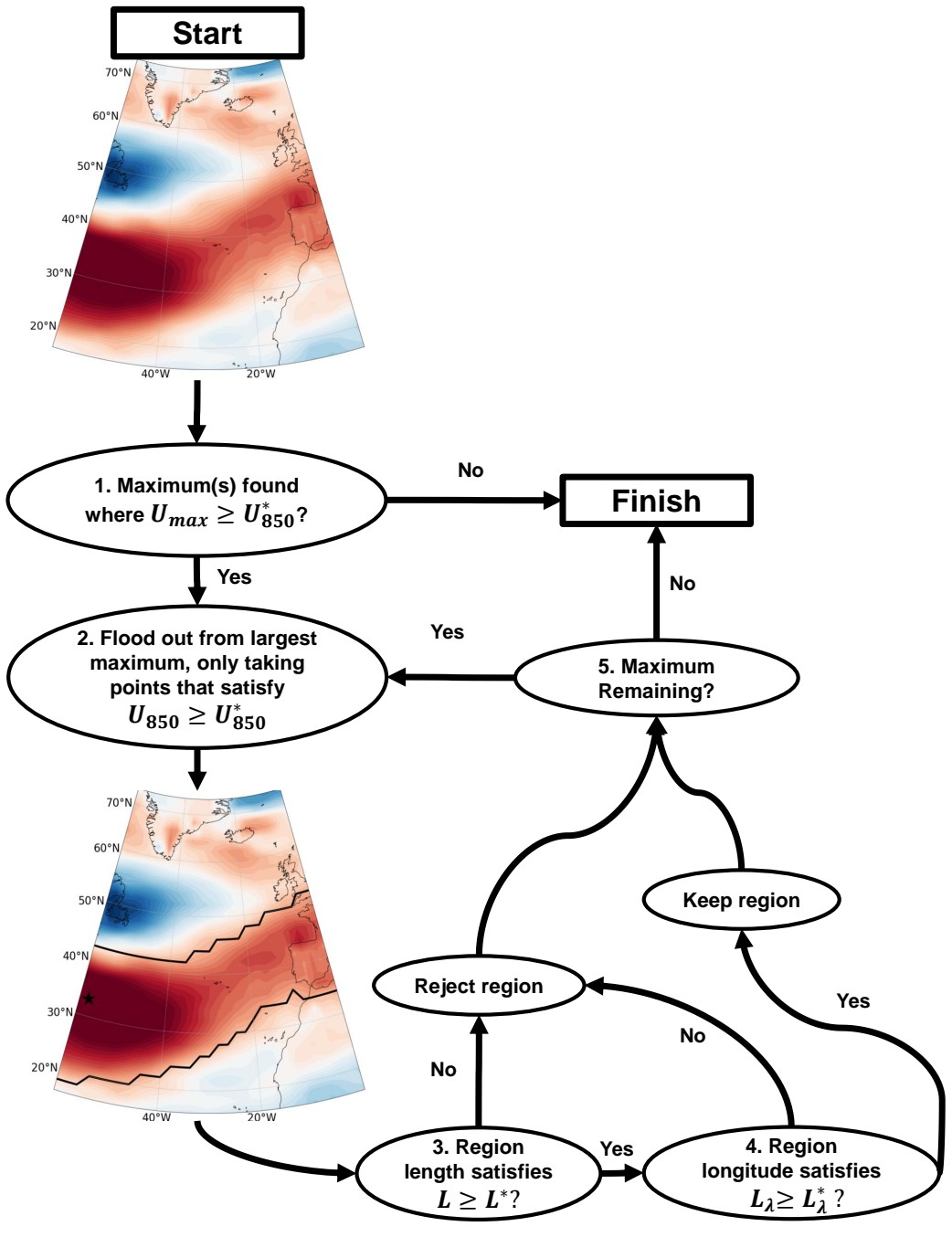

**Figure 1.** Algorithm for identification of Eddy-Driven Jet Objects (EDJOs). In the map, the black star is the seed point and the black contour is the EDJO found from the seed point.

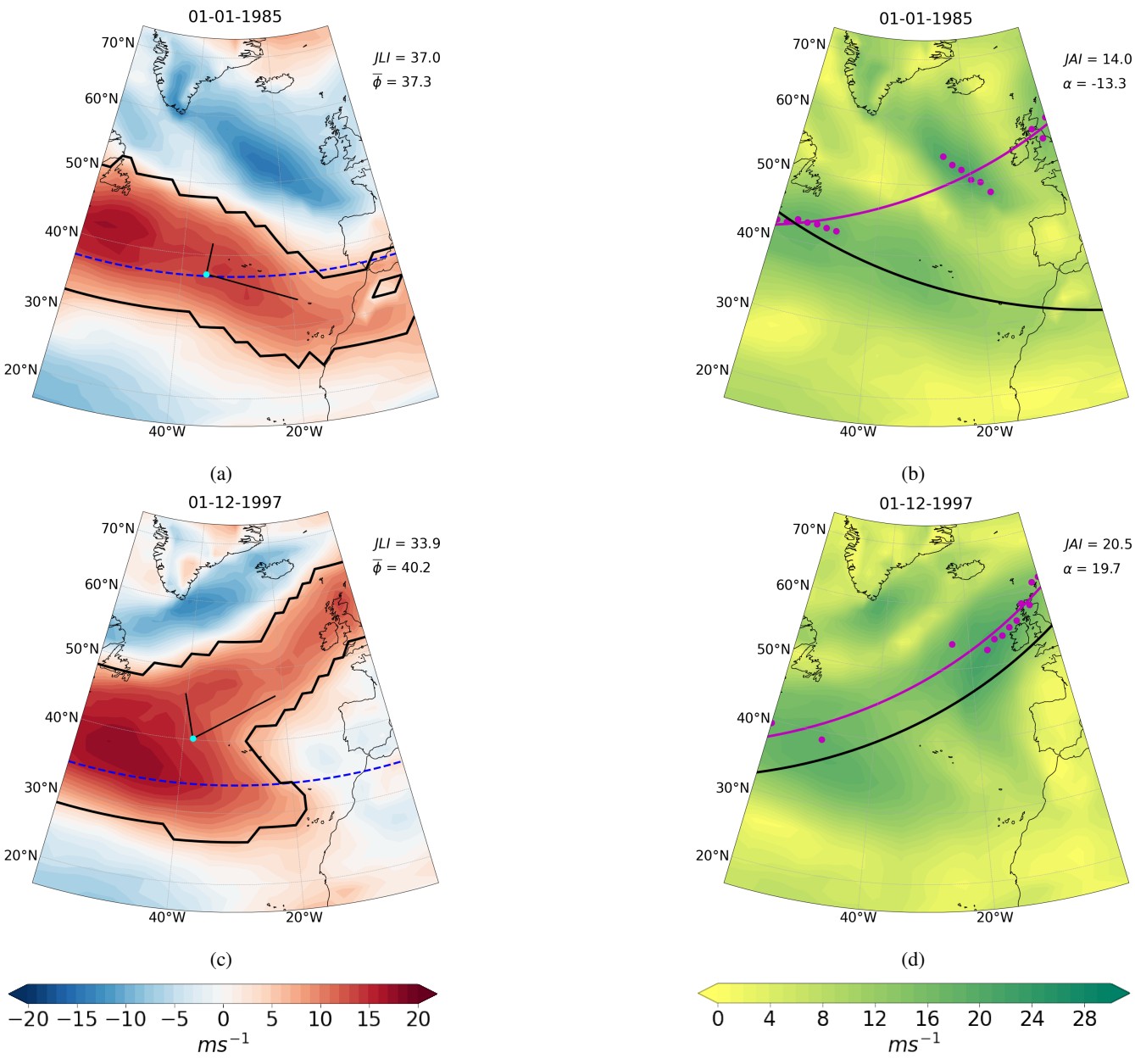

**Figure 2.** $U_{850}$ (left) and wind speed (right) for two different days (rows). The dashed line denotes the JLI. The black star denotes the location of the seed point and the light blue dot is the position of $(\overline{\lambda}, \overline{\phi})$ the centre of mass. In (b) and (d), the magenta circles are the maxima at each meridian and the magenta line is the result of linear regression fitted to those maxima following Messori and Caballero (2015). The JAI in (b) and (d) is calculated from the end points of the magenta line and the solid black line is the tilt given by $\alpha$ which comes from the black lines coming from the centre of mass in (a) and (c). Values of the indices for the respective methods are given in the top right of each panel.

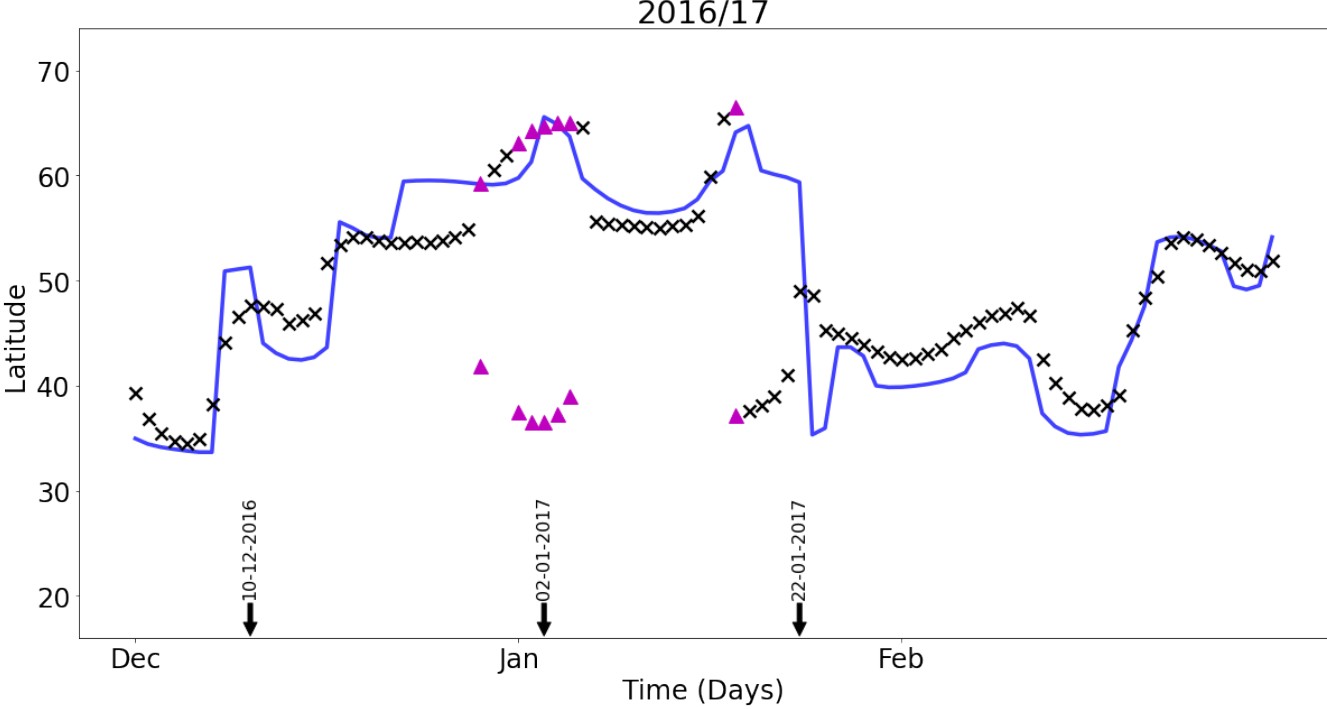

**Figure 3.** Time series of $\overline{\phi}$ (black crosses) and the JLI (blue line) for the boreal winter 2016/17. Pink triangles represent days with two EDJOs. The black arrows indicate the starting date for the consecutive day cases shown in Figure 4.

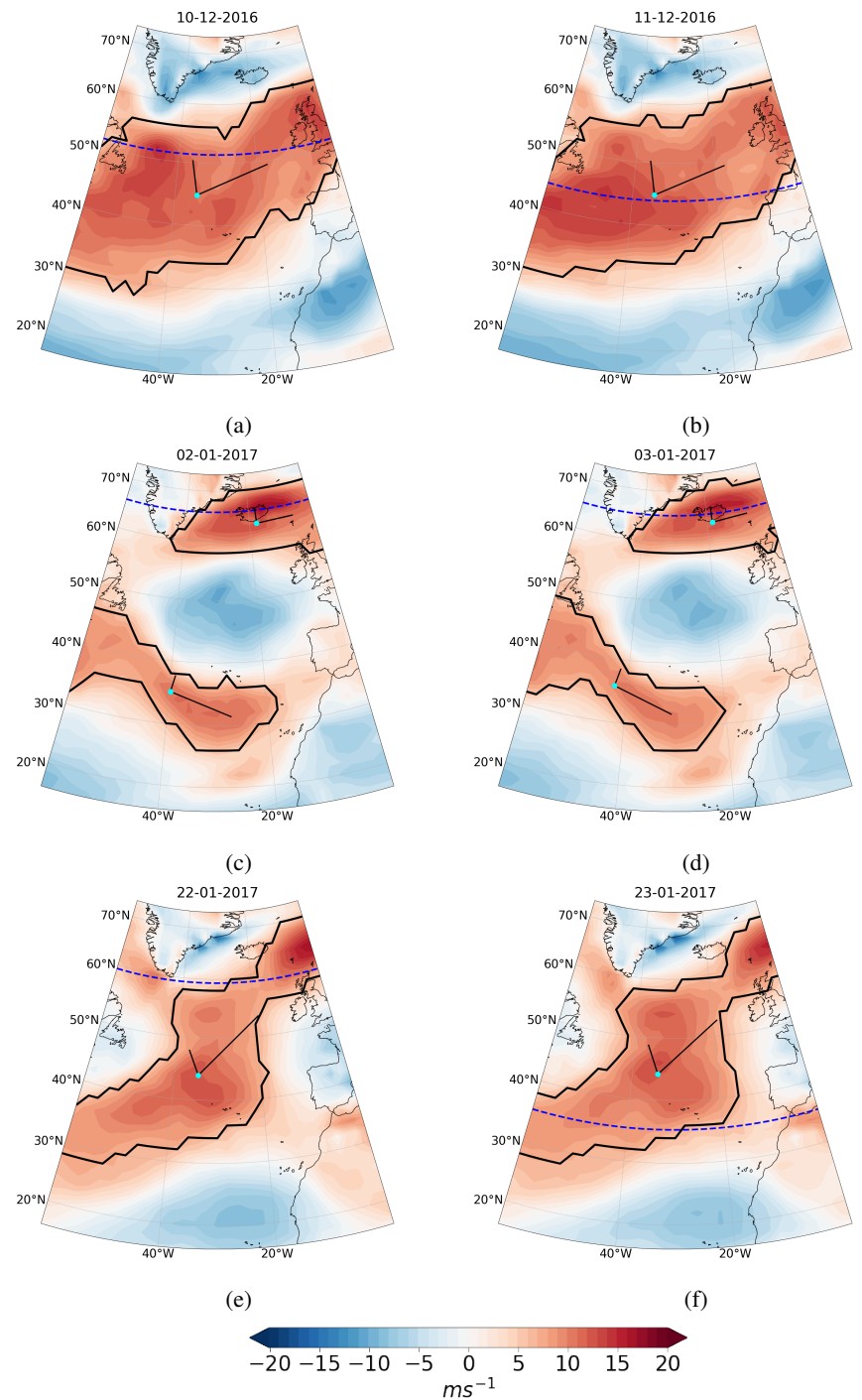

**Figure 4.** $U_{850}$ for example days in winter 2016/17 taken from Figure 3. Dashed dark blue lines denotes JLI value on each day. Solid contour(s) denote EDJO(s). Light blue dots denote the centre of mass $(\overline{\lambda}, \overline{\phi})$ of the EDJO with the longer black line defining the major axis and the shorter the minor axis.

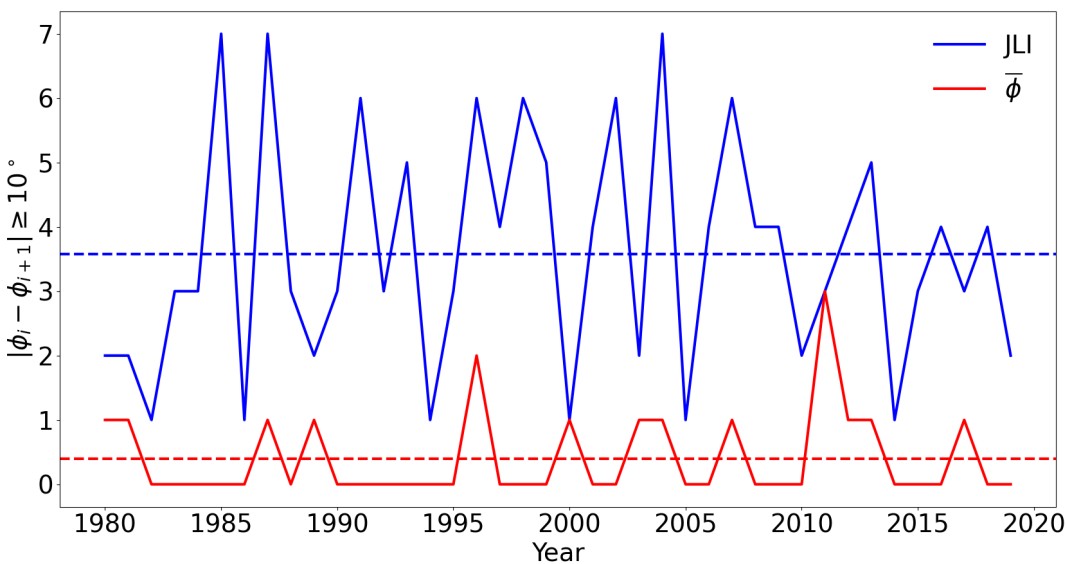

**Figure 5.** Frequency per winter of large shifts ($\geq 10°$) in jet latitude between consecutive days. Blue shows JLI shifts and red shows $\overline{\phi}$ shifts. The dashed lines show the average frequency per winter over the entire period. Calculations only account for days with one EDJO.

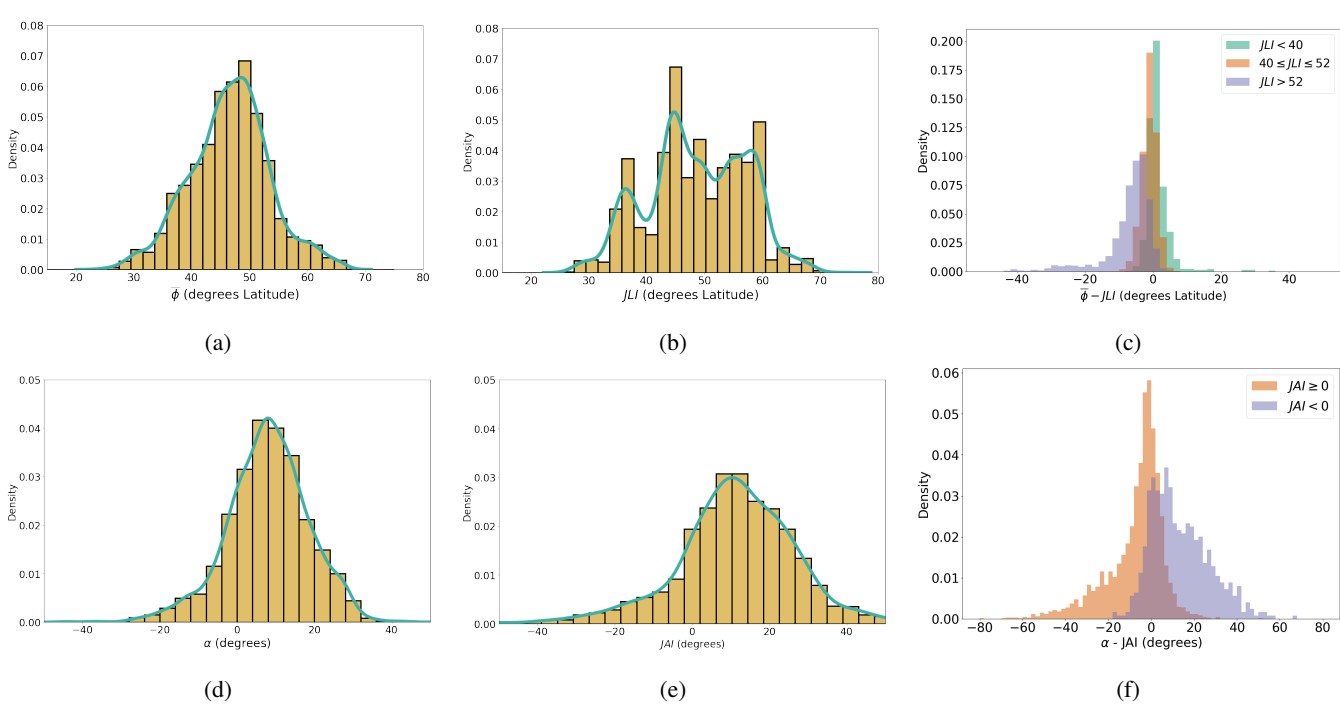

**Figure 6.** Distributions of all winter daily (a) $\overline{\phi}$, (b) JLI, (c) $\overline{\phi} - JLI$, (d) $\alpha$, (e) $JAI$ and (f) $\alpha - JAI$. In (c) $\overline{\phi} - JLI$ is coloured according to the three JLI modes and in (f) $\alpha - JAI$ is coloured based on the sign of $JAI$. Bin size is $2°$.

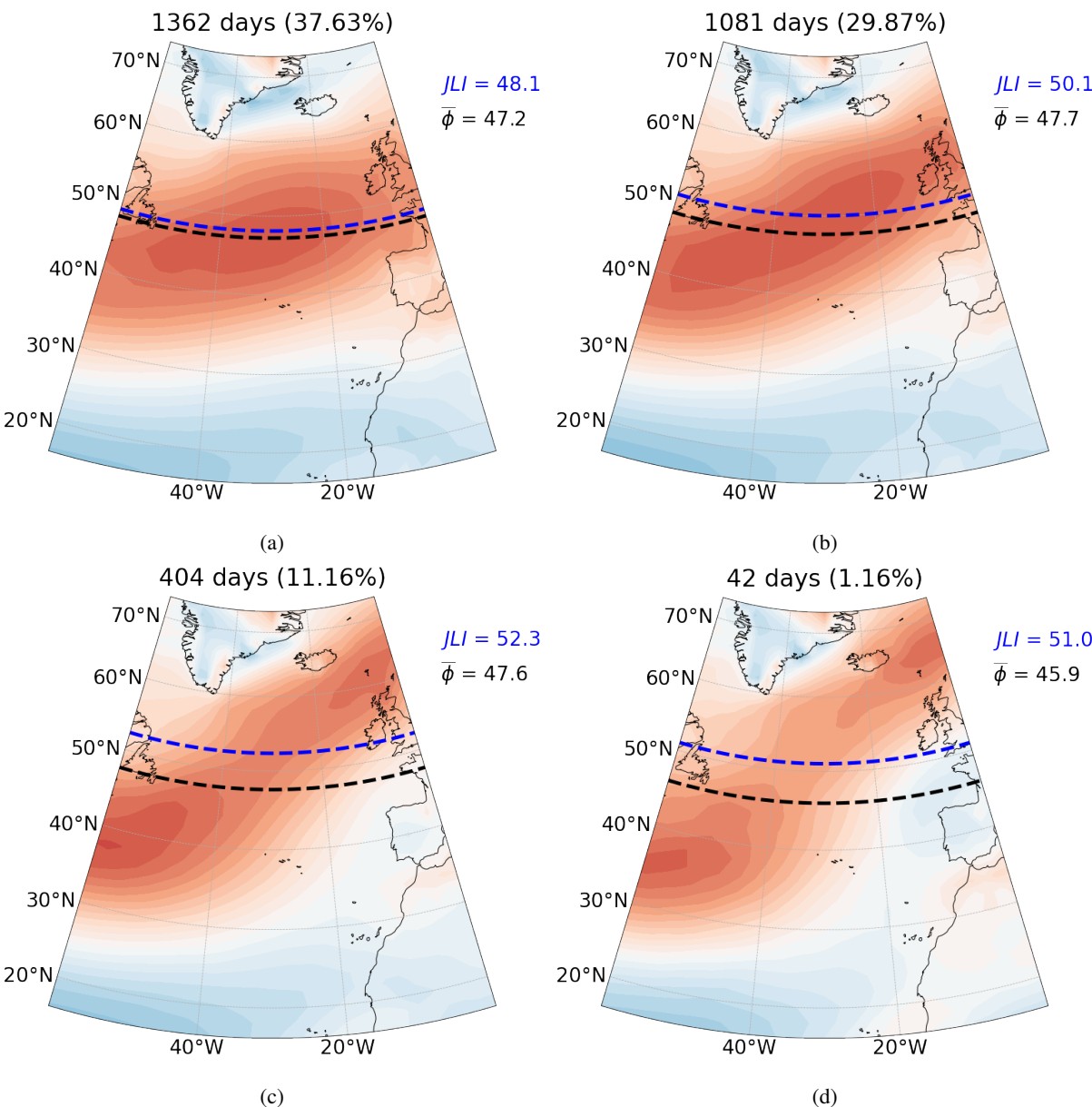

**Figure 7.** Zonal wind $U_{850}$ composites for positive $\alpha$ where (a) $0 \leq \alpha < 10$, (b) $10 \leq \alpha < 20$, (c) $20 \leq \alpha < 30$ and (d) $\alpha \geq 30$. Dashed lines represent the mean EDJ position for $\overline{\phi}$ (black) and JLI (blue). The number of samples in each composite is given in the header. Similar behaviour is seen for negative $\alpha$ (not shown).

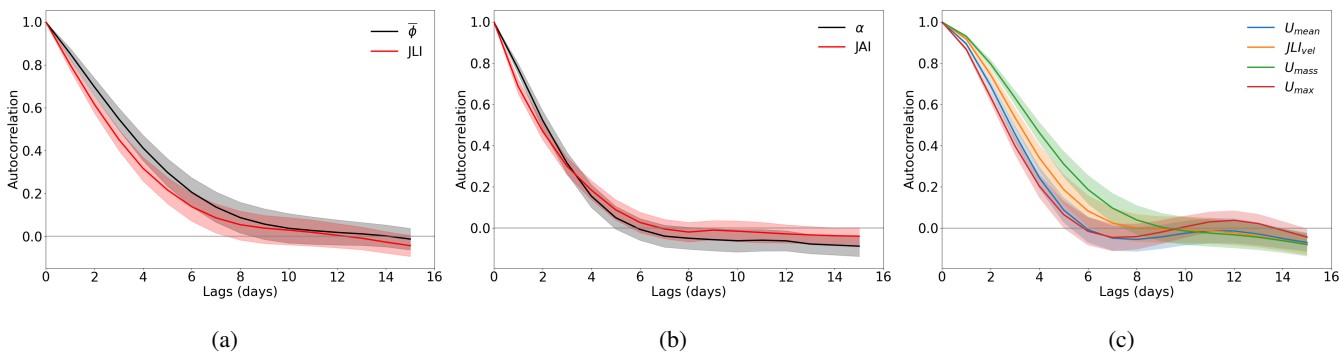

**Figure 8.** Lagged autocorrelation function for (a) $\overline{\phi}$ (black) and JLI (red), (b) $\alpha$ (black) and JAI (red), and (c) $U_{\mathrm{mean}}$, $U_{\mathrm{max}}$, $U_{\mathrm{mass}}$ and JLI$_{\mathrm{vel}}$. Shading in each plot represents two standard errors from the mean.

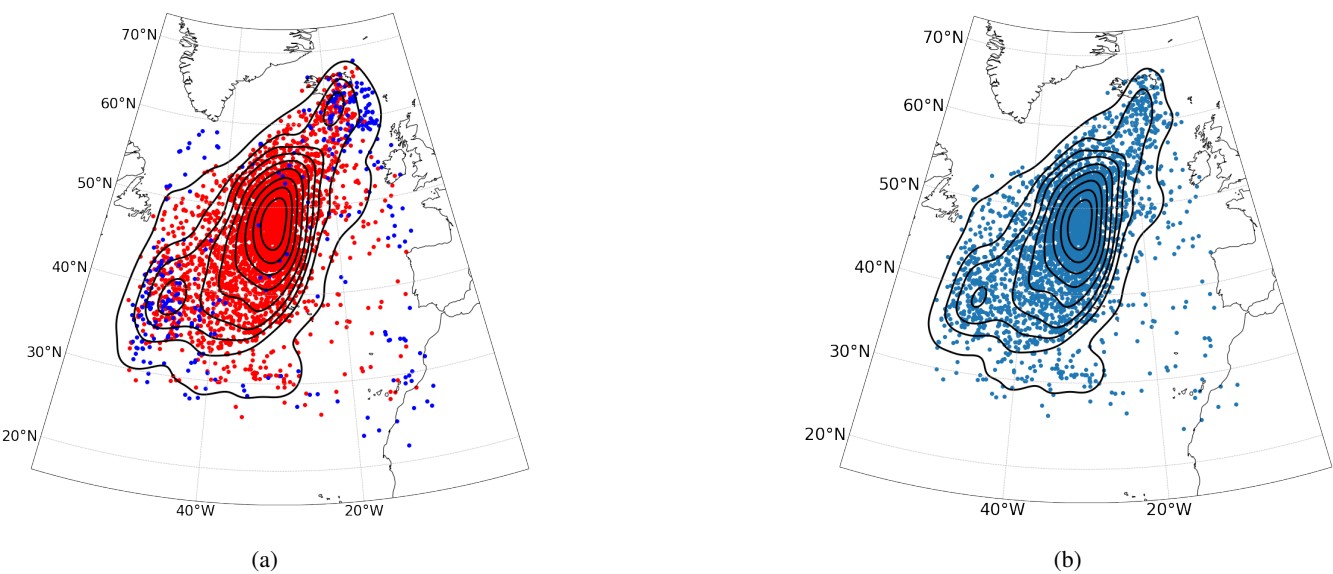

**Figure 9.** Distributions of EDJO centre of masses $(\overline{\lambda}, \overline{\phi})$ for (a) all EDJOs and (b) the largest $U_{\mathrm{mass}}$ EDJO on each day. Colours in (a) represent whether one (red) or two (blue) EDJOs are identified on a given day. Black contours show a two-dimensional KDE of the distribution.

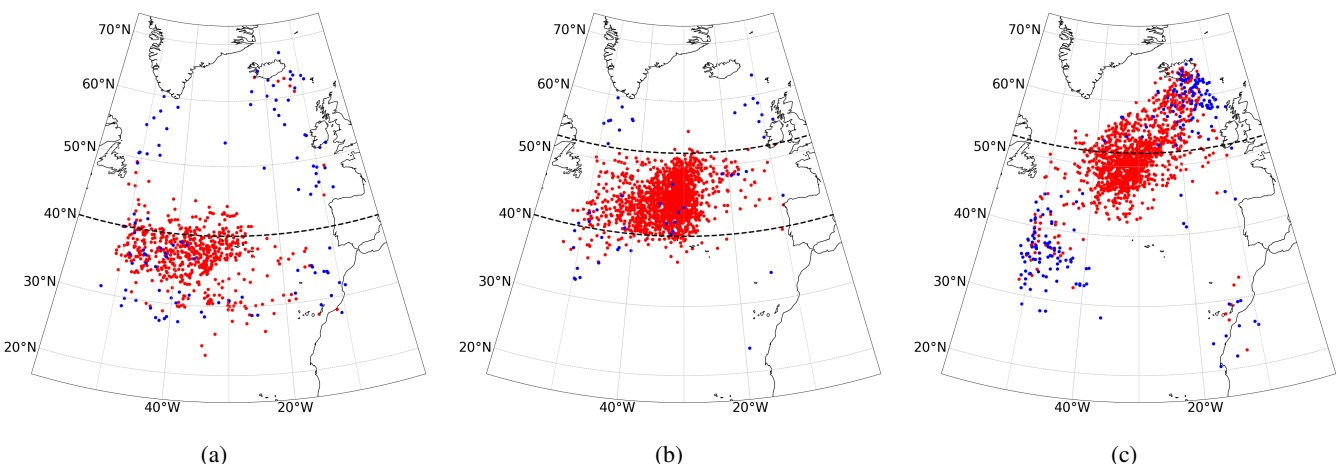

**Figure 10.** Distributions of the EDJO centre of mass, partitioned into the Southern (a), Central (b) and Northern (c) modes of the JLI. Colours represent whether one (red) or two (blue) EDJOs are identified on a given day. Black horizontal dashed lines indicate the latitudes used to define the JLI modes using the same bins as in Figure 6(c).

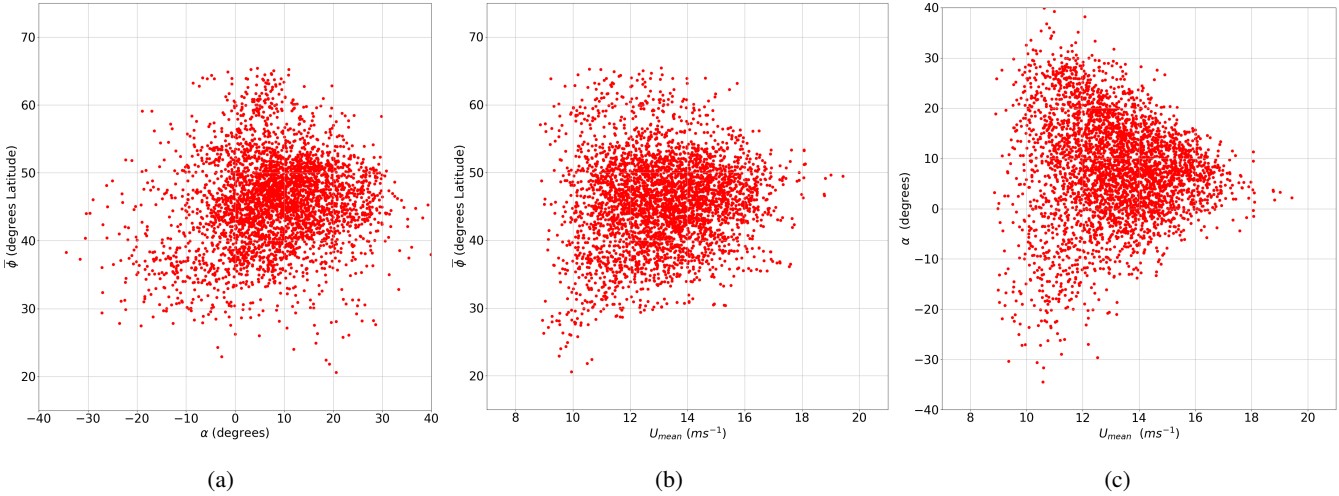

**Figure 11.** Scatterplots of all winter daily (a) $\overline{\phi}$ vs $\alpha$, (b) $\overline{\phi}$ vs $U_{\mathrm{mean}}$ and (c) $\alpha$ vs $U_{\mathrm{mean}}$.

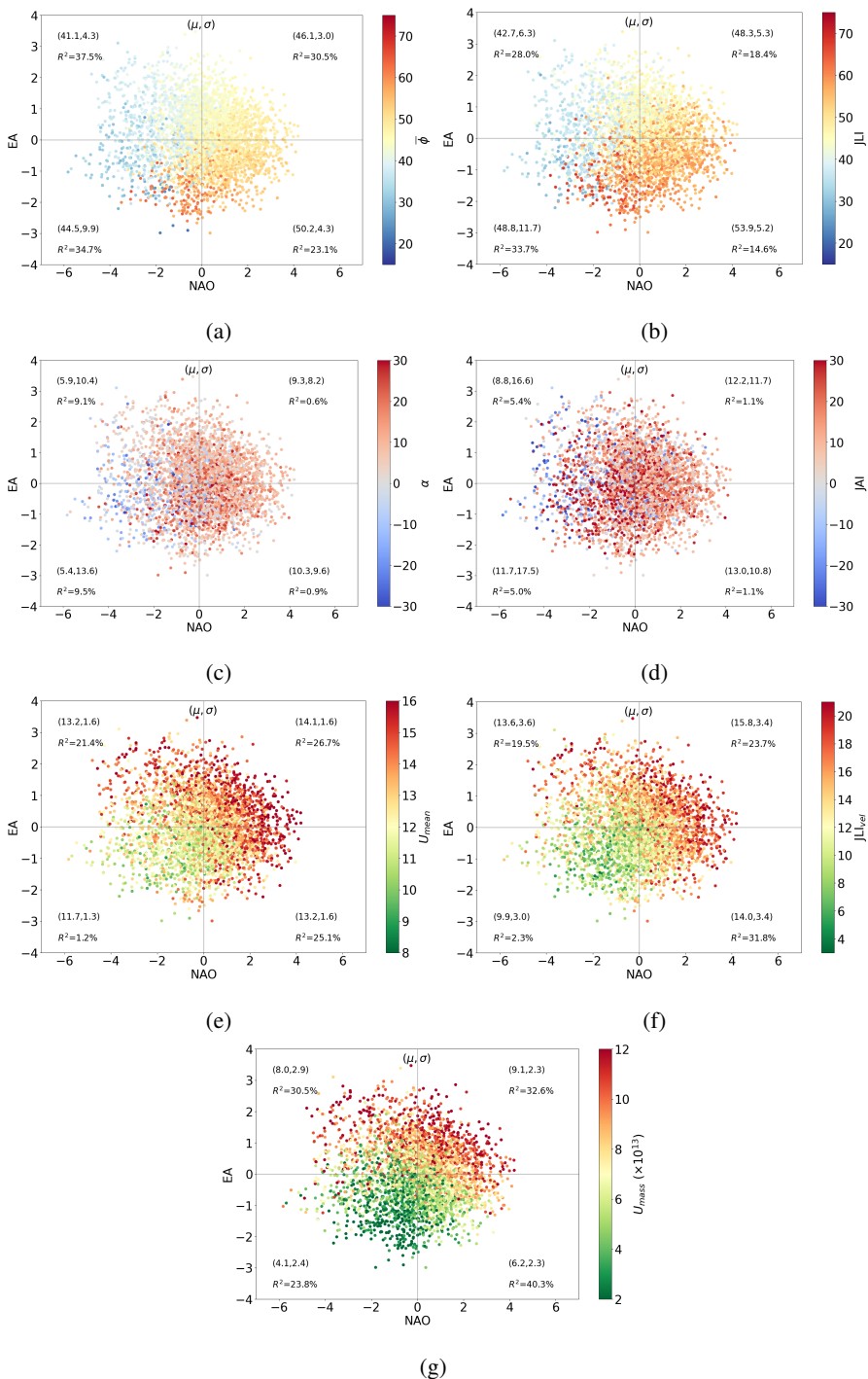

**Figure 12.** Scatterplots of all winter daily EA vs. NAO indices coloured by (a) $\overline{\phi}$, (b) JLI, (c) $\alpha$, (d) JAI, (e) $U_{\text{mean}}$, (f) JLI$_{\text{vel}}$ and (g) $U_{\text{mass}}$ for the EDJO with the largest mass. Note (e) and (f) are shown on different scales and $U_{\text{mean}}$ is bounded from below by $U_{850}^*$. Values of the mean $\mu$ and the standard deviation $\sigma$ for each quadrant is given in brackets and the variance explained $(R^2)$.

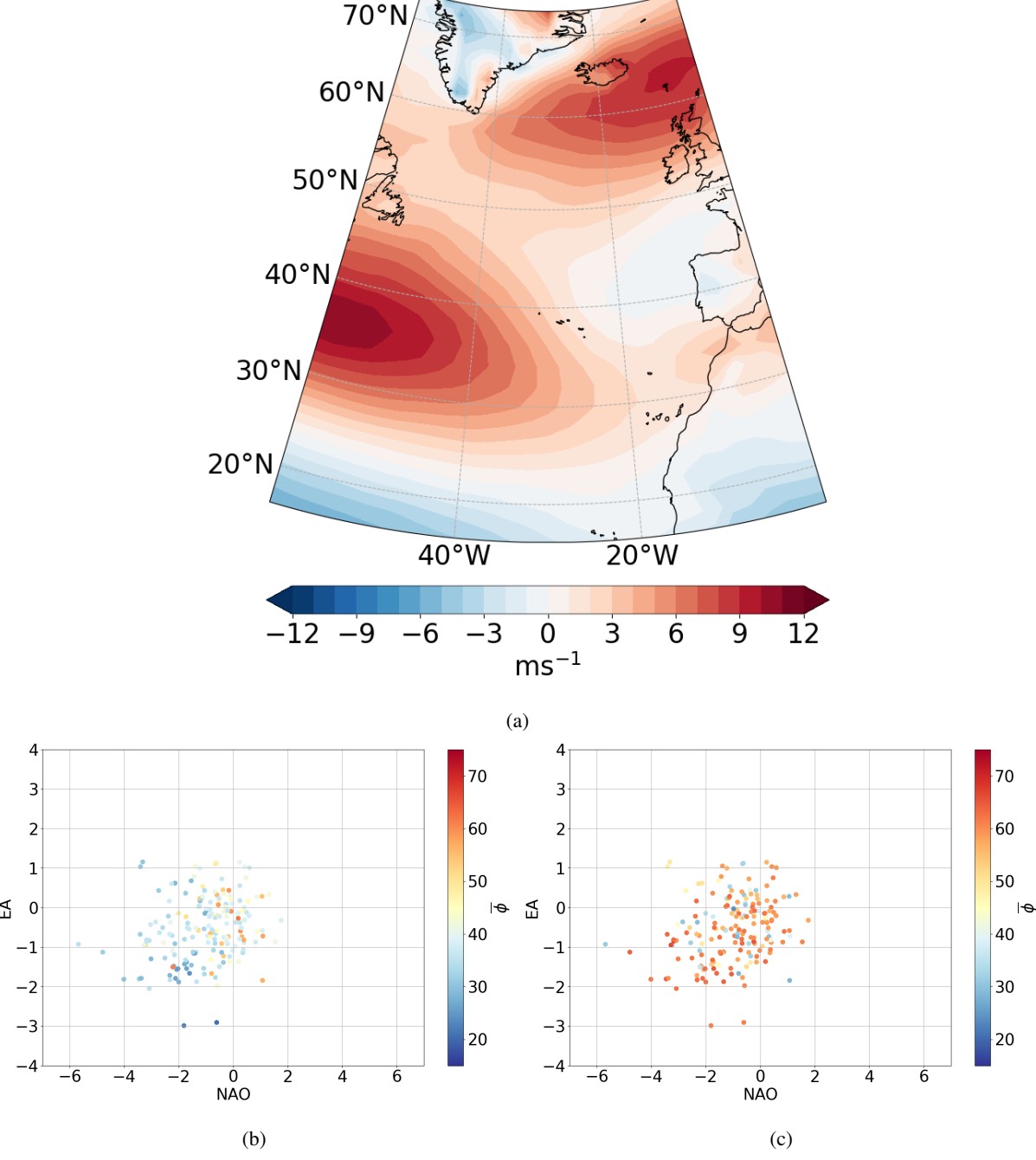

(a)

(b)

(c)

**Figure 13.** (a) Composite $U_{850}^*$ field for winter days with two EDJOs. This represents 4.9% of DJF days. (b, c) Scatterplots of all winter daily EA vs. NAO indices coloured by $\overline{\phi}$ for days with two EDJOs. (b) Shows the EDJO with the largest $U_{mass}$ and (c) shows the EDJO with the smaller $U_{mass}$.

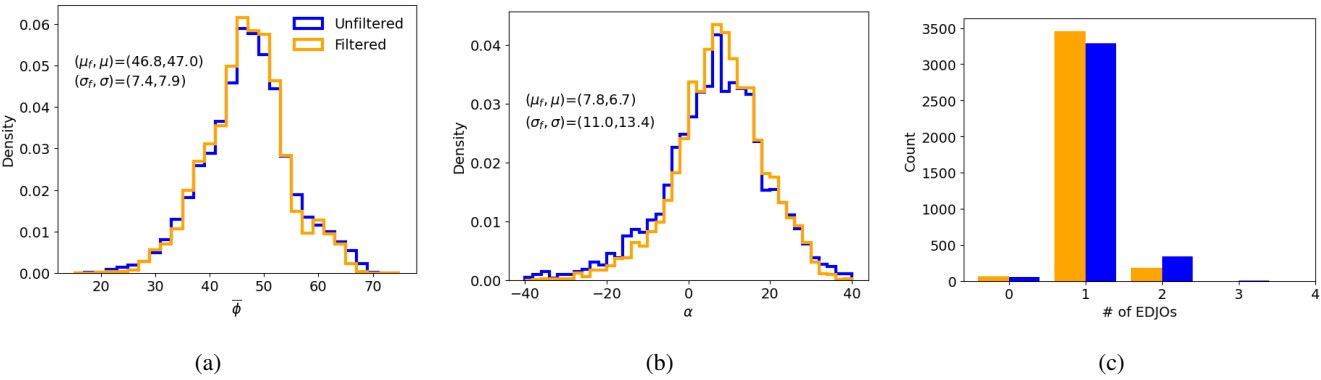

**Figure A1.** Distributions of $\overline{\phi}$ (a), $\alpha$ (b) and (c) the number of EDJOs identified for unfiltered (blue) and filtered (orange) $U^*_{850}$ data. The mean ($\mu$) and the standard deviation ($\sigma$) for each distribution are given in (a) and (c). The subscript $f$ denotes results based on filtered data.

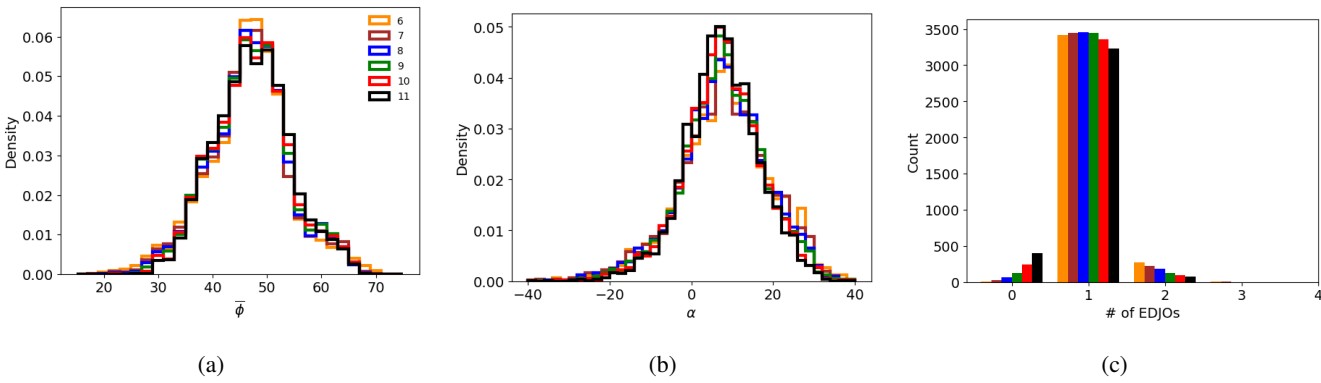

**Figure A2.** Distributions of $\overline{\phi}$ (a), $\alpha$ (b) and (c) the frequency of days with different numbers of EDJOs for a range of $U^*_{850}$ values from $6\,\mathrm{ms}^{-1}$ to $11\,\mathrm{ms}^{-1}$, with colours following the legend in panel (a).

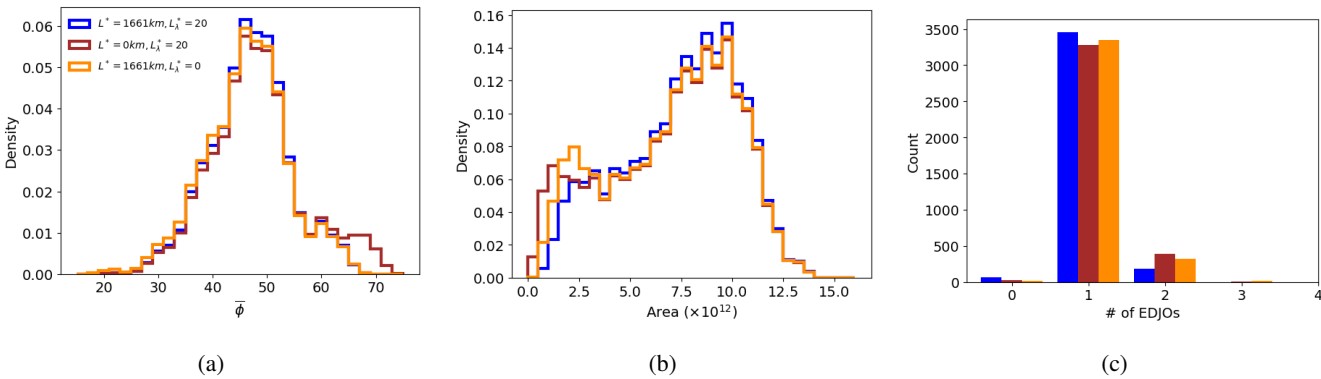

**Figure A3.** Distributions of $\overline{\phi}$ (a), Area (b) and the frequency of days with different numbers of EDJOs (c) with the inclusion of the $L^*$ and $L^*_\lambda$ checks (blue), just $L^*_\lambda$ (red) and just $L^*$ (orange).