# Peer review of "A new characterization of the North Atlantic eddy-driven jet using 2-dimensional moment analysis"

_EGUsphere, 2024_

## Author Comment (AC2)

**Reviewer Response**

Jacob Perez

March 2024

The reviewer comments are in black, and the author responses are in blue.

**Reviewer 1**

In the manuscript "A new characterization of the North Atlantic eddy-driven jet using 2-dimensional moment analysis" the authors suggest a more robust re-definition of two widely used jet metrics for the North Atlantic, jet latitude and tilt. The motivation, method and results are generally clearly presented, with only some caveats in the comments.

I still cannot recommend the manuscript in its current form for publication. First and foremost, I find it hard to regard their method to be a significant step beyond the current the state-of-the-art. In essence the authors so far only provide a technically more robust definition of jet latitude and tilt, but they do not address the conceptual problems associated with this approach (usage of low-level winds influenced by boundary layer processes, fronts, and orography; usage of low-frequency data to describe synoptic-scale dynamics; their jet latitude is still a zonal mean over a tilted storm track). Interestingly, the authors voice some of the same criticism over their predecessors, but then repeat the criticised choices in their method without further discussion. The resulting method is certainly more robust than its predecessors in a technical sense, it just remains unclear to what extent that constitutes a significant improvement if the main problems are of a conceptual nature. At the same time, conceptually more robust jet definitions would be available (cf. major comments A-C for details and references).

Having said that, I think the suggested method has potential for conceptual improvements beyond the state-of-the-art, but they are unfortunately not utilised so far. For example, the authors replicate existing jet latitude and tilt diagnostics, but do not explore how (combinations of) other moments might yield (more/other/complimentarily) useful diagnostics. May be some of combinations of moments describe jet waviness, waveguideability, or sharpness, in a useful and conceptually novel way? Further, while the authors in principle do allow for several EDJOs per time step, they then discard that additional information for most of their analyses (cf. also major comments C and D).

In summary, I think the authors need to...

(a) become much more precise in their criticism of previous studies given where their method actually can offer an improvement relative to the current state-of-the-art,

(b) properly acknowledge that alternative, conceptually more robust jet definitions would be available, and

(c) offer some conceptual improvement over previous North Atlantic jet metrics

... before this manuscript is suitable for publication.

C. Spensberger

The authors thank Dr. Spensberger for taking the time to carefully consider our manuscript. We have taken on board the suggestions for improving the manuscript detailed below, including addressing some of the reviewer's concerns about methodological choices. We also recognize there are other methods for jet identification, including some of the reviewer's own work, and we do not wish to present our approach as the only method worth using or indeed claim that it is the most superior method for all purposes. As such, we have extended the discussion of other studies on jet identification and softened the language in the conclusions about our approach. However, we strongly feel there is value in diversity and that the Woollings et al. (2010) method, which we build directly upon, has become arguably the most widely used method in the literature for identifying the North Atlantic jet. We agree with reviewer 2 who suggested that it should be up to the wider community to decide whether or not our approach offers sufficient advances on current methods to be useful.

We have not pursued a metric for jet waviness because we are aware of the issues with existing diagnostics (Geen et al., 2023) and did not want to add another metric to the mix. However, we have added more information to the manuscript about multiple EDJO days, thereby showcasing one of the features of the method.

**Major comments**

**Comment A** L34-35: I disagree. In my view, the main limiting factor for the reliability and interpretability is the attempt itself to capture daily North Atlantic jet state by a characteristic latitude and tilt based on time-filtered data. Time-filtering obscures physical processes, and trying to capture the widely meandering North Atlantic jet by these two metrics is necessarily an extreme simplification. For applications where such extreme simplification is permissible, simply considering, for example, the latitude of maximum wind might very well be sufficient. Replacing the latitude of maximum winds by a weighted mean latitude does not solve any of the conceptual limitations of this approach.

**Response:** Following Woollings et al. (2010), we apply time filtering because we aim to characterise the EDJ variations on longer timescales than individual weather systems. However, here we show that the overall outcomes of the method are largely insensitive to this time filtering. Figure 1 shows the distributions of $\overline{\phi}$, $U_{\mathrm{mean}}$, $\alpha$, $U_{\mathrm{mass}}$ and EDJO area when using filtered and unfiltered $U_{850}$ data. There are very minor differences in the distributions of $\overline{\phi}$ and $\alpha$ (Figure 1a and c), with each distribution having very similar $\mu$ and $\sigma$. The distributions of $U_{\mathrm{mean}}$ (Figure 1b) show the largest difference, with stronger EDJOs found in the unfiltered data than in the filtered data, as expected when removing a smoothing filter.

We also include the distributions of $U_{\mathrm{mass}}$ and the area of each EDJO, shown in Figures 1d and e. The difference in $U_{\mathrm{mass}}$ follows the difference in $U_{\mathrm{mean}}$, where there is a higher occurrence of larger objects in the unfiltered data. However, from the area distribution we can see that this is just due to the changes in strength and not a result of finding larger objects, as once again

there is little difference between the filtered and unfiltered distributions of the area. The last dependence on filtering we test is the number of EDJOs found (Figure 1e). Here we see that the number of zero days is almost identical in both cases, and there are some modest differences in the number of single and two-object days. The difference between single- and two-object days could be attributable to the stronger winds in the unfiltered data, as this would allow for longer-lasting EDJOs.

Overall, we conclude that time filtering has little impact on EDJO detection and the resulting statistics of the parameters, with the main difference being a higher mean jet strength. We have added an Appendix in the manuscript titled "Robustness of EDJO Identification" where this discussion is included.

[Figure]

Figure 1: Distributions of $\overline{\phi}$ (a), $U_{\mathrm{mean}}$ (b), $\alpha$ (c), $U_{\mathrm{mass}}$ (d), Area (e) and the number of Eddy-Driven Jet Objects (f) for time filtered $U_{850}$ (blue) and unfiltered $U_{850}$ (orange) based on ERA5 data over 1979-2020. Values of the mean ($\mu$) and standard deviation ($\sigma$) are given in each panel, with a subscript $f$ referring to the value for the filtered data. For panel (f), the values are the count for each respective bin.

**Comment B** L22-24 versus L55: In the same vein, it seems weird that the authors criticise Woollings et al. (2010) following White et al. (2019), but then themselves use the same low-level winds to define their jet indexes. The jet indices suggested in this manuscript will be equally much affected by the Greenland orography as the jet indices in Woollings et al. (2010), although the orographic influence will likely manifest itself in a different way. More generally than White et al. (2019), Spensberger et al. (2023) recently showed that low-level winds are not a good indicator for the presence of eddy-driven jets. Even just a latitude threshold would be a better choice than the low-level wind for separating subtropical from eddy-driven jets. Spensberger et al. (2023) suggest defining eddy-driven jets as jets occurring below the isentropic level of 335K.

**Response**: White et al. (2019) conclude that the northerly peak of the JLI distribution may not represent a 'regime' of the EDJ but rather contamination by orographically-forced low level jets. In referencing White et al. (2019), we have interpreted the results as evidence of the issues associated with data dimensionality reduction and selecting a single maximum as jet position. Although White et al. (2019) underscores the importance of careful interpretation, it does not necessarily discourage the use of low-level winds altogether. We also highlight that many studies use low-level winds as a diagnostic for EDJ (e.g., Barriopedro et al., 2022; Marcheggiani et al., 2023), especially in climate model studies. While we recognise the sophistication of the identification method in Spensberger et al. (2017), we note that this method requires calculation of high-frequency potential vorticity. High frequency climate model output is often limited (e.g. from CMIP6) and comes on coarse resolution horizontal and vertical grids, which can lead to inaccuracies when calculating derivatives. Therefore, there are merits in having a method that requires only one parameter at a single level ($U_{850}$) as input. We believe that this simpler data requirement is one reason why the Woollings et al. (2010) approach has become so widely adopted.

We do not agree with the reviewer's assertion that 'the jet indices in this manuscript will be equally affected by the Greenland orography as the jet indices in Woollings et al. (2010)'. To motivate this, Figure 2 shows the effect of the two length checks we impose on the EDJOs ($L^*$ and $L_\lambda^*$, as described in Section 2.2 of the manuscript). In Figures 2a and 2e it can be seen that the inclusion of the length checks substantially reduces the identification of small EDJOs at far northern latitudes near Greenland. It is highly likely that many of the removed EDJOs are linked to tip jets, which are intense - and can hence be detected by Woollings et al. (2010) - but tend to be of smaller scale. Hence, the inclusion of the length checks in our algorithm reduces the risk of detecting Greenland tip jets, and the effect is therefore not similar to that on the JLI.

[Figure]

Figure 2: Distributions of $\overline{\phi}$ (a), $U_{\mathrm{mean}}$ (b), $\alpha$ (c), $U_{\mathrm{mass}}$ (d), Area (e) and the number of Eddy-Driven Jet Objects (f) for different $L^*$ and $L_\lambda^*$. Blue uses $L^*$ and $L_\lambda^*$ as in the main text. Red uses $L_\lambda^*$ but neglects $L^*$. Yellow uses $L^*$ but neglects $L_\lambda^*$.

**Comment C** Section 2.2/Figure 1: I do not agree that the jet identification method is particularly simple, nor does it seem particularly robust. I am certainly biased here to some extent, but to me the Spensberger et al. (2017) definition of jet axes as "well-defined lines of wind maxima" (using basically one threshold to mathematically define the term "well-defined") even seems conceptually simpler than jet objects iteratively defined by the flow chart in Figure 1. In terms of implementation, Spensberger et al. (2017) is surely somewhat more complex, but then both a reference implementation (Spensberger, 2021) and the actual detections for ERA5 (Spensberger et al., 2023) are publicly available. The same is true for JETPAC, a conceptually similar jet definition by Manney et al. (2014) that has been used in a series of studies on jet and storm track dynamics.

**Response:** We acknowledge the existence of other identification methods focused on upper tropospheric variables and have included more discussion of these in the Introduction, as this was lacking before. The simplicity of our approach comes in part from the need for only a single input variable at a single pressure level. As noted before, output from climate models is often limited but daily $U_{850}$ is frequently available. The other methods mentioned above require output fields at high spatial and temporal resolution which may not be consistently available from climate models.

Regarding the robustness of the definition, the iterative definition of jet objects (repeating steps 2-4) will make the identified jet objects sensitive to small variations in the input wind field. I have experimented with similar approaches, and specifically the topological property of what is connected above the chosen wind speed threshold is essentially random. A more robust way to

segment the wind field into several jet objects would be to use a watershed algorithm, that is iterating through the wind field not by "flooding" based on seed points, but rather iterating through the grid points in order of decreasing wind speed. Then, for every point in isolation, one can use well-defined and explicit criteria to decide whether a grid point is (a) a new jet object, (b) an extension of an existing jet object, or (c) a grid point either merging or separating two jet objects.

As context and illustration for this comment, I would also point out the preprint of Auestad et al. (2024): any spatio-temporal filtering in the jet definition muddies the otherwise clear relation between the jet and latent heating. Only when considering the instantaneous meanders of jets is the latent heat release clearly concentrated on the warm side of the jet, the side where one would physically expect it.

**Response:** Thanks for highlighting this interesting study. However, the goal of our study is not to examine the role of diabatic processes for jet behaviour. As stated before, we agree other methods might be better suited for answering specific science questions and we think diversity in approaches can be valuable. We also refer the reviewer back to Major Comment 1 where we show the time filtering does not strongly affect the overall results.

With regard to the sensitivity 'to small variations in the input wind field', we address this by presenting the results for a range of $U_{850}^*$ values, shown in Figure 3. Here, it can be seen from Figures 3a and c that both $\bar{\phi}$ and $\alpha$ are largely insensitive to the choice of $U_{850}^*$. There are changes in $U_{\mathrm{mean}}$, as expected, since the distribution is bounded by $U_{850}^*$, however, there is some agreement around the mean of $U_{850}^*$ from 6-9 m s$^{-1}$. Some differences are also seen in the distributions of area and $U_{\mathrm{mass}}$, which is also to be expected because the size of the EDJOs is related to $U_{850}^*$. Finally, the distribution of the number of EDJOs also shows little difference for a value of $U_{850}^*$ between 6-9 m s$^{-1}$. Higher values of $U_{850}^*$, lead to a decrease in days with one EDJO and an increase in days with zero. We have added these sensitivity tests into the manuscript under a new Appendix with the heading 'Robustness of EDJO identification'.

The reviewer's suggestion to adopt a watershed algorithm is interesting, but after considering this we realised that to differentiate between (a), (b) and (c) would require specific choices and criteria that may introduce further undesirable dependencies.

In short: using either the Spensberger et al. (2017) jet axes or the Manney et al. (2014) JETPAC would yield more reliable and interpretable results than the new method proposed here.

**Response:** We believe the method to be reliable based on the results of different tests given in the response to Main comments A, B, and D. The intepretability of results we also believe to be clear as we are only defining very commonly used characteristics of the EDJ.

**Comment D** L186-187: I don't understand that choice. The authors seem to thus discard one of the stated main benefits of their method: the ability to potentially detect several jet objects per time step.

**Response**: We agree, the choice of leaving out the two object days when comparing to the JLI was to assess the days with a single EDJ definition as is done with the JLI. We have now included more results

[Figure]

Figure 3: Distributions of $\overline{\phi}$ (a), $U_{\mathrm{mean}}$ (b), $\alpha$ (c), $U_{\mathrm{mass}}$ (d), Area (e) and the number of Eddy-Driven Jet Objects (f) for a range of $U^*_{850}$ values shown in the legend.

in the manuscript regarding the two object days under and new section with heading 'Two object days'.

**Comment E** There are several large-area figures which do not contain large amounts of information (in particular Figs. 3, 5, 6, 10 and 11; to some degree also Figs. 2 and 4). I am sure some of the information therein could be combined and conveyed in a more compact way. The discussion of some Figures remains quite superficial, indicating that some Figures might more be illustrating side remarks than the core idea that the authors want to convey. Such Figures might with benefit not be shown or moved to a supplement.

**Response:** We believe these figures are important for demonstrating the performance of the algorithm and for providing context to the main conclusions of the manuscript. However, we agree that the discussion around these figures can be improved, and hence more detail has been added.

**Minor Comments**

**L14:** Correction: Spensberger et al. (2017) detect both STJs and EDJs. Spensberger et al. (2023) thoroughly investigate how to separate STJs from EDJs, and thus implicitly offer an EDJ-only detection algorithm.

**Response:** Adjusted in the manuscript.

**L68-70:** How does the flooding handle object mergers? I.e. what happens if two seed points are connected at a level above the minimum wind speed? Does the stronger EDJO always win? If so, there would be more robust ways of separating jet objects (cf. major comment C).

**Response:** In the event that two or more seed points lie within the same connected $U_{850}$ contour, the results are unaffected since it is treated as one object and the moments are calculated for the entire object irrespective of the position of the seed point. Unique EDJOs are only identified when seed points lie within a $U_{850}$ contour that meets the length checks.

**L81-85:** This seems to confirm my hunch in L68-70.

**Response:** See previous response.

**L185:** Note that these statistics will depend heavily on the filtering of the input field. Without filtering, there'd be several EDJs (and probably also EDJOs) almost all the time during winter.

**Response:** See response to Main comment A.

**L204-205:** With that finding, it must be the mass-weighted definition of the jet latitude that makes the decisive difference, rather than the option for multiple EDJOs per time step?

**Response:** Yes, the mass weighting would cause the difference in the distributions. The sentence has been extended to include this.

**Fig 8:** Beyond a tilt angle of 20 degrees, neither jet latitude metric seems to do justice to the composite average wind pattern. And the composite will certainly be cleaner than the individual time steps from which it is compiled.

**Response:** The purpose of this plot was to highlight the structure of the circulation when northerly jets are defined in the JLI, not whether the respective metric is better at defining at latitude on tilted jet days.

**L225-226:** I am not able to follow this argument. Which search are the authors referring to, and why would that cause noise/ a lower autocorrelation?

**Response:** Here we are referring to the large latitudinal jumps in the JLI, which would reduce the ACF.

**L248-249:** As in main point D: curious that the authors chose to move this result featuring a novel aspect of their method in the supplement and focus the main manuscript on replicating known analyses.

**Response:** See response to main point D.

**L277-288:** Given the authors so far only offer a technical improvement, I feel these conclusions are way too strong. Being as bold as the authors, I could equally validly claim that the results of Spensberger et al. (2023) and preprint of Auestad et al. (2024) suggest that also results obtained with the new method introduced here would better be reconsidered using a more robust jet definition such as theirs. I would of course never do that ;-)

**Response:** Phrasing of the manuscript and the conclusions have been toned down accordingly.

**Reviewer 2**

Review of "A new characterization of the North Atlantic eddy-driven jet using 2-dimensional moment analysis" by Jacob Perez et al.

This study presents a new method for characterising variability of the North Atlantic eddy-driven jet on 10-day timescales. The method identifies 'eddy-driven jet objects' based on a threshold U850 value, and employs spatial moments of the objects to define jet latitude and tilt diagnostics. This overcomes some well-documented shortcomings with other widely-used jet diagnostics (notably by identifying multiple jet objects when the flow is highly distorted, thereby providing a cleaner interpretation on such days) whilst remaining relatively simple to calculate.

The manuscript and figures are clear and well-written. It is a methods paper which does not contain any new science results as far as I can tell, but the results presented provide an informative, albeit light-touch, comparison to other jet latitude and tilt diagnostics from the literature. I note that RC1 is concerned about whether this new method provides any significant improvements over the previous methods. My view is that this decision can be left to the wider community by the extent to which the new method is adopted. I do, however, have several minor issues with the results as currently presented which I would like to see addressed.

We thank the reviewer for reading and providing feedback on the manuscript. We agree with the reviewer that the intention of this work was to introduce another method using low-level winds to define characteristic features of the EDJ. The responses to the reviewers comments and the changes for the revised manuscript are given below.

**Major Comments**

**Comment 1** Given (as noted in CC1) you place strong emphasis on the fact the distribution of phi is unimodal and state that this 'casts doubt on the previous interpretations of the trimodal distribution as evidence for regime behaviour', I am also surprised you do not explore further the fact that the spatial distribution of EDJO centre of masses in Fig 6 remains trimodal. Please provide further interpretation of this result (for example, it could be informative to explore how the centre of mass distribution (Fig 6b) varies for days when the JLI is in its N, M or S position), or else describe more clearly how you reach your conclusion regarding the evidence for regime behaviour.

**Response:** The weak multimodality apparent in the centre of mass map in Figure 6a is mainly due to the occurrence of two object days, when objects are typically found near Iceland and the east of the USA (compare Figure 6a and 6b in the manuscript). Furthermore, when integrating in longitude, $\overline{\phi}$ is unimodal (Figure 7a). Nevertheless, as suggested by the reviewer, we have partitioned the centre of mass distribution into three JLI bins shown below in Figure 4. For the southern and central JLI bins, most of the EDJO centres of mass lie close to or within the JLI bin with the exception of secondary EDJOs (blue points). However, for the northern JLI bin, this breaks down and a substantial proportion of the centres of mass lie outside the JLI bin. This underlines some of the issues we highlight in the manuscript with the JLI detecting northerly states. There is also a difference in the number of two EDJO days within each partition, the highest percentage

occurring in the northern JLI bin (2.8% days) followed by the southern (1.4% days) and then the central (0.68% days).

We thank the reviewer for this comment, as it has added valuable analysis and provided further evidence for our conclusions. The discussion of the centre of mass distribution has been extended to include these details.

[Figure]

Figure 4: Spatial distribution of the centres of mass for the EDJOs, categorized according to the peaks in the JLI. Here, red points represent days characterized by a singular EDJO, whereas blue markers denote days with two EDJOs. Black dashed lines define the latitude boundaries that define the JLI latitudinal regimes: the southern regime (a) is defined by latitudes less than 40°, the northern regime (c) by latitudes exceeding 52°, and the central regime occupies the region between the two.

**Comment 2** Given this is a methods paper, I would like to see a more detailed analysis of the robustness of the methodology to the choices made. As some examples, you use 10-day LP filtered winds as input and a threshold of $U_{850}^*=8$m/s. What's the justification for these choices, and to what extent are the results sensitive to them? The reason I ask is that whilst the 10-day filter is used in the 'standard' Woollings et al method, the results obtained there are largely insensitive to this choice. I suspect the EDJO objects, in contrast, may break up into many smaller objects if less or no temporal smoothing was applied.

**Response:** The choice of filtering is done because it is one of the most popular choice of filter from the literature and from testing it made negligible difference to our overall results when choosing to filter or not (see response to Comment A from Reviewer 1). The choice of 8ms$^{-1}$ for $U_{850}^*$ was motivated by the identification of jet events in Section 2 paragraph 4 of Woollings et al. (2010), where the same threshold is used to detect jet events. Please see the response to main comments A and C for Reviewer 1 for the sensitivity of these choices and we have also added an Appendix to the revised manuscript titled 'Robustness of EDJO identification'.

**Minor Comments**

**L3:** Please provide a confidence interval for the skewness to convince the reader it is robustly negative (also in the main text).

**Response:** Thank you for suggesting this addition. The 95% confidence interval for the skewness is $(p_{2.5}, p_{97.5}) = (-0.15, 0.02)$ based on a bootstrapping of the $\overline{\phi}$ distribution with 10,000 samples. This is now included in the manuscript.

**L5:** What do you mean by 'less noise'? Please be more specific. This sentence suggests that the method casts doubt on the stated interpretation because it is less noisy. Is this really what you mean?

**Response:** By less noise we mean to say a higher persistence. This has been changed in the manuscript.

**L17 (first half):** Please provide the specific refs which give this interpretation.

**Response:** References have been added to the manuscript.

**L55:** I realize that you use the same domain as Woollings et al deliberately, but it is striking (Figs 2 and 4) that most of your EDJOs are cut-off, particularly at the western boundary. Would you expect greatly different results if your domain extended further west?

**Response:** Extending the domain further west has some influence on our results, this is shown in Figure 5 where four different domains have been tested. When the domains are extended further west, we can see little change in the distribution of $\overline{\phi}$ (Figure 5a) with a slight increase in the number of equatorward points. This is consistent with Figure 2 in Woollings et al. (2010), where a high density of grid points with zonal wind larger than $8\,\mathrm{m\,s^{-1}}$ is found in this region. The distribution of $U_{\mathrm{mean}}$ (Figure 5b), with the strength of the EDJOs in the western domains slightly weaker than the other domains. This is due to the average of an EDJO that is also larger due to the use of the larger domain, which is evidenced by the distributions of $U_{\mathrm{mass}}$ and Area (Figure 5d and e). There is little to no change in the distribution of $\alpha$.

**L75:** What distance is used if the major axis intersects the object boundary several times?

**Response:** If the major axis intersects the object boundary twice, the distance is calculated from the intersection on the outer boundary.

**L201:** 'median value' $\rightarrow$ 'median difference'.

**Response:** Changed.

**213:** Figure 9 is only mentioned in passing, and can be removed without losing any of the messages of the paper.

**Response** Figure 9 has been moved into the supplementary.

[Figure]

Figure 5: Distributions of $\overline{\phi}$ (a), $U_{\mathrm{mean}}$ (b), $\alpha$ (c), $U_{\mathrm{mass}}$ (d), Area (e) and the number of Eddy-Driven Jet Objects (f) for different North Atlantic domains.

**L246:** To my mind, the fact that the variation is NAO/EA space is smoother for phi and alpha than JLI and JAI is a key result of the paper which should be mentioned in the conclusions (and possibly abstract). However, is it not the case that the total variance is larger for JLI and JAI than phi and alpha (estimated from Fig 7)? Therefore, having smaller variance in each quadrant does not, by itself, evidence this point. Please provide a cleaner analysis of the fraction of variance explained.

**Response:** Thank you for this comment we agree that this is an interesting part of the work and should be mentioned more. The abstract and conclusions of the manuscript have been updated, including this information, and the figures have been changed to include the variance explained ($R^2$) for each of the quadrants for each of the metrics. The original discussion has been extended to include this information.

**L264:** It would be good to see some analysis of the jet width, which would add a more novel component to your results.

**Response:** The length and width of the EDJOs don't produce any interesting results as they scale by the overall strength of the EDJO. The information is somewhat captured in $U_{\mathrm{mass}}$ as this is a measure of the size of an EDJO. We have included more information on $U_{\mathrm{mass}}$ in the revised manuscript, in the 'Large-Scale modes of Variability' section.

**L287:** I'd question that the method only has three tuneable parameters (presumably $U_{850}^*$, $L^*$ and $L_\lambda^*$?). Specifically, the choice of low-pass filter and domain of study surely have a big impact of the results.

**Response:** Please see the response to main comment A from reviewer 1 with regards to sensitivity of the EDJOs to time filtering and also the response to your comment on L55 about the choice of domain. Here we have added the clause 'for a given domain the method has....'.

**Fig 2 caption:** Which dates are shown?

**Response:** Dates have been added to these figures in the revised manuscript.

**Fig 10:** Please add an indication of sampling uncertainty to the autocorrelation lines to indicate where the differences between the different diagnostics are robust.

**Response:** The plot has been changed to include shading for two standard errors for the sampling uncertainty.

**References**

Barriopedro, D., Ayarzagüena, B., García-Burgos, M., and García-Herrera, R. (2022). A multi-parametric perspective of the North Atlantic eddy-driven jet. *Climate Dynamics*, 61(1–2):375–397.

Geen, R., Thomson, S. I., Screen, J. A., Blackport, R., Lewis, N. T., Mudhar, R., Seviour, W. J. M., and Vallis, G. K. (2023). An explanation for the metric dependence of the midlatitude jet-waviness change in response to polar warming. *Geophysical Research Letters*, 50(21).

Manney, G. L., Hegglin, M. I., Daffer, W. H., Schwartz, M. J., Santee, M. L., and Pawson, S. (2014). Climatology of Upper Tropospheric–Lower Stratospheric (UTLS) Jets and Tropopauses in MERRA. *Journal of Climate*, 27(9):3248–3271.

Marcheggiani, A., Robson, J., Monerie, P., Bracegirdle, T. J., and Smith, D. (2023). Decadal Predictability of the North Atlantic Eddy-Driven Jet in Winter. *Geophysical Research Letters*, 50(8).

Spensberger, C. (2021). Dynlib: A library of diagnostics, feature detection algorithms, plotting and convenience functions for dynamic meteorology.

Spensberger, C., Li, C., and Spengler, T. (2023). Linking Instantaneous and Climatological Perspectives on Eddy-Driven and Subtropical Jets. *Journal of Climate*, 36(24):8525–8537.

Spensberger, C., Spengler, T., and Li, C. (2017). Upper-Tropospheric Jet Axis Detection and Application to the Boreal Winter 2013/14. *Monthly Weather Review*, 145(6):2363–2374.

White, R. H., Hilgenbrink, C., and Sheshadri, A. (2019). The Importance of Greenland in Setting the Northern Preferred Position of the North Atlantic Eddy-Driven Jet. *Geophysical Research Letters*, 46(23):14126–14134.

Woollings, T., Hannachi, A., and Hoskins, B. (2010). Variability of the North Atlantic eddy-driven jet stream: Variability of the North Atlantic Jet Stream. *Quarterly Journal of the Royal Meteorological Society*, 136(649):856–868.

---

## Author Response (AR2)

**Author response to the Editor: Perez et al., A new characterization of the North Atlantic eddy-driven jet using 2-dimensional moment analysis, submitted to Weather and Climate Dynamics**

**July 2024**

Editor comments in black. Author response in blue.

I'd like to thank the authors and both of the reviewers for their work in improving this paper. Reviewer 2 has a few minor outstanding points to be addressed but Reviewer 1 has declined to provide a recommendation. Following the advice of Reviewer 2, the decision as to whether this is a sufficient advance over existing methods can be left for future users to decide. The method is clearly described and tested in the revised manuscript, which both reviewers noted is considerably improved. This paper is a useful contribution to the literature, so long as it is presented in the context of the diversity of metrics available which emphasise different aspects of the flow.

We thank the Editor and both reviewers for reading the revised manuscript and to the Editor and Reviewer 2 for offering further suggestions for improvements. We have implemented these as detailed below. We are disappointed with the attitude of Clemens Spensberger who appears unwilling to engage constructively with the peer review process despite our efforts to take on board his suggestions.

The paper could be accepted following consideration of the following minor suggestions:

1. The enhanced introduction is useful but could go further to highlight that both types of approach have advantages and applications. As potential examples, the lower-tropospheric metrics provide an integrated view of the flow, connecting well to patterns such as the NAO as shown (which ultimately explain a large fraction of the variance) and have also been found to relate clearly to the zonal momentum budget. Equally, the upper tropospheric metrics can reveal details associated with individual synoptic systems, and relate better to diabatic processes as mentioned.

    Thank you for the comment, the following sentences have been added to the introduction:

    "Upper tropospheric jet metrics are particularly useful for connecting the jet with synoptic systems (e.g., Spensberger et al., 2017) and elucidating the influence of diabatic processes on the jet (e.g., Auestad et al., 2024) "

    On lines 23-24.

"Lower tropospheric metrics have been shown to have close links to large-scale modes of variability such as the North Atlantic Oscillation (e.g., Barnes and Hartmann, 2010; Woollings et al., 2010) and the zonal momentum budget in the mid-latitudes (e.g., Simpson et al., 2014)."

On lines 31-33.

2. The effects of time filtering have been a particular topic of interest. The additional sensitivity tests have certainly helped here. I think it's also possible that the nature of the new method is somewhat scale-selective, with a focus on the larger scales. So although no explicit spatial filtering is applied, some may be implicit, hence the relative lack of sensitivity to time filtering. This could be considered and discussed if relevant.

Thank you for highlighting this interesting point. We do have explicit spatial filtering incorporated through $L^*$ and $L^*_\lambda$, so this likely reduces the sensitivity to time filtering by removing small EDJOs that may be detected in unfiltered data. It is less clear to us what role there is for implicit spatial filtering. In the absence of explicit spatial filtering, any implicit filtering would be connected to the choice of $U^*_{850}$. If $U^*_{850}$ is sufficiently small to only identify large EDJOs in unfiltered data, then it could act as an implicit spatial filter. However, this would be offset against identifying smaller regions of weak westerlies. For higher $U^*_{850}$, the EDJO spatial scale would get progressively smaller and would identify small areas of localised strong wind in unfiltered data, which would not be removed in the absence of explicit spatial filters. However, we expect that in practice the relative lack of sensitivity to time filtering is most strongly influenced explicit spatial filtering through $L^*$ and $L^*_\lambda$.

The following sentence has been added to address this:

"We note this result is likely to depend on the inclusion of explicit spatial filtering through $L^*$ and $L^*_\lambda$, which remove small EDJOs that might be more frequently detected in unfiltered data."

On lines 393-394.

3. The role of the length scale in reducing the impact of Greenland tip jets is clear in the revised manuscript. It might be worth mentioning as a caveat another common limitation of lower-tropospheric metrics is that different modelling systems have different approaches for treating below-ground regions of the 850hPa surface, so there can be some sensitivity to this.

Thank you for your comment. We have added the following sentences to the conclusions:

"One limitation affecting lower tropospheric jet metrics is that different approaches are used for interpolating below the surface over high topography, e.g. Greenland, which can affect the input wind fields. Therefore, care is required to check the influence of, e.g. missing data over Greenland for lower tropospheric metrics."

On lines 371-374.

4. The authors should check that statements they have made about the Spensberger method are correct. In general, it might help to highlight positive aspects of other indices, such as that by Barriopedro et al, as well as aspects that they are trying to improve on.

Statements regarding the method of Spensberger et al. (2017) have been checked and corrected. We have also added the following sentence regarding the metrics proposed by Barriopedro:

"Barriopedro et al. (2022) also extend the measures of the EDJ to better characterise its variability by introducing measures of longitudinal position and sharpness; however, the implementation

of their measures relies on similar underpinning assumptions as Woollings et al. (2010) and Messori and Caballero (2015)."

On lines 59-62.

5. The comments made by Reviewer 2 should, of course, be addressed.

The changes suggested by reviewer 2 have been implemented in the manuscript. The descriptive statistics given on line 3 in the abstract have been removed as we agree that it's not clear how these contrast with the JLI. The sentence on line 3 now reads:

"the distribution of the daily winter $\overline{\phi}$ is unimodal which is in contrast to the trimodal distribution of the daily Jet Latitude Index (JLI) described by Woollings et al. (2010)."

The change from **noise** to **higher persistence** has now been made on line 5. We apologise for not doing this the first time.

We have corrected line 21 from **vertical** to **quasi-horizontal** when discussing the work of Spensberger et al. (2017).

Thank you for this comment, we agree that this is an added benefit of the method and the following sentence has been added to lines 71-73 of the manuscript:

"Further, issues with the identification of orographic features are mitigated by incorporating a minimum EDJ length, which is shown to reduce the occurrence of jets located at high northern latitudes."

The change to line 83 has been made.

**References**

Auestad, H., Spensberger, C., Marcheggiani, A., Ceppi, P., Spengler, T., and Woollings, T. (2024). Spatio-temporal filtering of jets obscures the reinforcement of baroclinicity by latent heating. *EGUsphere*, 2024:1–19.

Barnes, E. A. and Hartmann, D. L. (2010). Dynamical Feedbacks and the Persistence of the NAO. *Journal of the Atmospheric Sciences*, 67(3):851–865.

Barriopedro, D., Ayarzagüena, B., García-Burgos, M., and García-Herrera, R. (2022). A multi-parametric perspective of the North Atlantic eddy-driven jet. *Climate Dynamics*.

Messori, G. and Caballero, R. (2015). On double Rossby wave breaking in the North Atlantic. *Journal of Geophysical Research: Atmospheres*, 120(21):11,129–11,150.

Simpson, I. R., Shaw, T. A., and Seager, R. (2014). A Diagnosis of the Seasonally and Longitudinally Varying Midlatitude Circulation Response to Global Warming. *Journal of the Atmospheric Sciences*, 71(7):2489–2515.

Spensberger, C., Spengler, T., and Li, C. (2017). Upper-tropospheric jet axis detection and application to the boreal winter 2013/14. *Monthly Weather Review*, 145(6):2363 – 2374.

Woollings, T., Hannachi, A., and Hoskins, B. (2010). Variability of the North Atlantic eddy-driven jet stream. *Quarterly Journal of the Royal Meteorological Society*, 136(649):856–868.